# Design, Modeling, and Control of a Composite Tilt-Rotor Unmanned Aerial Vehicle

Zhuang Liang [1], Li Fan [2,*], Guangwei Wen [2] and Zhixiong Xu [3]

1   School of Aerospace Science and Technology, National University of Defense Technology, Changsha 410073, China; liangzhuang15@nudt.edu.cn
2   Huzhou Institute, Zhejiang University, Huzhou 313000, China; wengw@hizju.org
3   College of Control Science and Engineering, Zhejiang University, Hangzhou 310027, China; 22232005@zju.edu.cn
*   Correspondence: fanli77@zju.edu.cn

**Abstract:** Tilt-rotor unmanned aerial vehicles combine the advantages of multirotor and fixed-wing aircraft, offering features like rapid takeoff and landing, extended endurance, and wide flight conditions. This article provides a summary of the design, modeling, and control of a composite tilt-rotor. During modeling process, aerodynamic modeling was performed on the tilting and non-tilting parts based on the subcomponent modeling method, and CFD simulation analysis was conducted on the entire unmanned aerial vehicle to obtain its accurate aerodynamic characteristics. In the process of modeling the motor propeller, the reduction of motor thrust and torque due to forward flow and tilt angle velocity is thoroughly examined, which is usually ignored in most tilt UAV propeller models. In the controller design, this paper proposes a fusion ADRC control strategy suitable for vertical takeoff and landing of this type of tiltrotor. The control system framework is built using Simulink, and the control algorithm's efficiency has been verified through simulation testing. Through the proposed control scheme, it is possible for the composite tiltrotor unmanned aerial vehicle to smoothly transition between multirotor and fixed-wing flight modes.

**Keywords:** tilt-rotor; flight dynamics; fusion ADRC; control simulation

## 1. Introduction

In recent years, vertical takeoff and landing (VTOL) unmanned aerial vehicle (UAV) have enormous development potential in short distance transportation applications such as urban air transportation in the civilian field [1] and have attracted widespread attention from the global commercial market. The vertical takeoff and landing UAV does not require a long runway, nor does it require separate launch and retrieval devices, and has the characteristics of simple operation, maneuverability, and flexibility [2]. At present, the most common UAV are mainly multirotor and helicopter. Although these types of UAV have excellent performance during hovering and low speed forward flight, they have a common drawback: poor performance during high-speed forward flight. Although fixed wing aircraft avoid the problem of insufficient performance during forward flight, they have certain limitations on the minimum flight speed, and takeoff and landing depend on the airport. As the third type, the VTOL UAV combines rotating and fixed wings on a flight platform, this feature allows the aircraft to attain vertical takeoff and landing capabilities, in addition to facilitating efficient horizontal flight [3].

Vertical takeoff and landing aircraft can be divided into the following main categories according to their flight mode conversion methods: standard composite VTOL (dual-systems), tail-sitter, and tilt type (tilt-rotor, tilt-wing). The standard composite VTOL is the easiest to control, but the separate hover/forward flight propulsion system adds additional weight. The tail-sitter VTOL does not require an additional power tilting mechanism and has the smallest actuator group. However, due to the overall tilting of the fuselage, they

have significant uncertainty during the tilting process, and the impact of wind stroke during hovering and tilting is significant [4]. In comparison to tail-sitters, tilt-rotor aircraft offer greater control privileges and are easier to manage during hover, but the inclusion of the extra tilt mechanism leads to an augmentation in the overall complexity of the system. Research on tiltrotor aircraft primarily emphasizes the modeling of characteristics during the transition process and the development of stable control methods for the transition period.

There is a large amount of references on the development of tilt-rotor UAVs. Here, we present a comprehensive review of the reference concerning methodologies pertinent to the modeling and control aspects of tilt-rotor UAVs. In terms of tilt-rotor UAV modeling, Ducard et al. [5] developed a tilt-rotor aircraft with four motors that can tilt separately and performed a dynamic modeling of the target aircraft. In terms of power, the modeling and simulation of propeller force and torque are mainly considered, including the impact of the incoming flow speed of the motor propeller disc due to tilt and the angular velocity of the motor around the rotation center on the propeller modeling. Wang et al. [6] analyzed the aerodynamic characteristics of a large tilt-rotor aircraft during the mode conversion process based on the blade element method and adopts a component modeling method. Shamsheer et al. [7] modeled the flight of the tilt-wing aircraft in the optimization of the tilting wing VTOL takeoff trajectory. In the process of studying the dynamic characteristics, they used empirical formulas to model and analyze the wing stall characteristics during the tilting process. The momentum theory is used to model and analyze the power loss caused by the tilt of the propeller disk, and the normal component perpendicular to the propeller disk caused by the incident angle is considered.

In terms of control of tilting vertical take-off and landing UAVs, Li et al. [8] introduced a novel tilting three-rotor UAV design that combines the vertical flight attributes of a helicopter with the horizontal flight capabilities of a fixed-wing aircraft. The control logic employs fixed-wing and multirotor control modules to engage in various flight control phases of the aircraft, and a hybrid control of the two controllers is used in the transition stage. Liu et al. [9] proposed a new TRUA-V transition control method based on a multi-model adaptive method for the control problem of nonlinear controlled objects under the constraints of inclined corridors. Shen et al. [10] took a tilting three-rotor UAV as the research object and designed a full-mode flight control law for the tilting-rotor aircraft based on the incremental nonlinear dynamic inverse method, realizing the tilt-rotor aircraft from rotor mode to Smooth transition from fixed wing mode. Wang et al. [11] introduced an attitude controller utilizing Active Disturbance Rejection Control (ADRC) to address model uncertainty issues encountered during the tiltrotor transition process. Through simulation and experimental evaluations, it was demonstrated that the controller exhibits notable resistance to external disturbances and achieves precise control accuracy. Guillaume et al. formulated a nonlinear model predictive control for the control of a tilt-rotor with quadrotor synchronous tilting capabilities [12–14]. Yin et al. [15] proposed an adaptive control algorithm to eliminate approximation errors for attitude tracking control of tiltrotor quadcopters and developed a new M-CBDCS control architecture to adapt to the uncertainty of model parameters during flight. Liao et al. [16] proposed a hybrid HDO-STSMC control scheme for a new tiltrotor UAV and verified the feasibility. The feasibility and effectiveness of the proposed control architecture with hybrid functionality were validated through simulation. In terms of redundant control of tiltrotor, Mousaei et al. [17] introduced an instantaneously online multi-controller distribution technology that allows tilt-rotor quadcopters to make control unit adjustments when a failure occurs. In terms of intelligent control, Xu et al. [18] proposed a method to achieve hybrid UAV transition control by training a modeless, model-independent neural network controller.

This article introduces a tilt-rotor eVTOL, unlike other tiltrotor, our VTOL UAV is equipped with 6 rotors (2 tiltable propellers and 4 fixed propellers for vertical lift) to balance the different working conditions of the propellers in multirotor and fixed wing mode, and to meet the requirements of high efficiency, lightweight, and low cost, used for attitude

control and power generation in different flight modes. The modeling of flight dynamics and control simulations are presented. During the modeling process, we consider the loss of thrust and torque along the propeller axis in the presence of incoming flow. Factors such as aerodynamic interaction between propellers and wings were also considered. In addition, we conducted computational fluid dynamics simulations on for the aircraft to capture its full aerodynamic characteristics. For the design of flight controller, a fusion ADRC control strategy is proposed for this type of tiltrotor UAV, specifically for a smooth transition from vertical take-off to level flight or level flight to vertical landing. Flight simulations were conducted for testing the proposed scheme. The contributions of this article are as follows:

(1) Propose a distributed composite tilting aircraft that can stably and reliably perform the transition between hovering mode and horizontal flight, and adopt two power systems to better adapt to different flight conditions without using variable pitch propellers.

(2) Aerodynamic modeling uses a combination of numerical simulation and empirical simulation and uses the idea of component modeling to split the aerodynamic analysis of the entire machine into fixed components and tilting components, achieving low-cost and efficient modeling and simulation.

(3) A fused ADRC control architecture suitable for the target aircraft is proposed. The controller has certain anti-disturbance characteristics and can stabilize the aircraft under certain external disturbances and achieve smooth transition and flight in different modes.

The subsequent sections of this article are structured in the following manner: In the Section 2, the overall design including aerodynamic design, propulsion system, and a brief introduction of tilt-rotor mechanical design are provided. In Section 3, we introduced the dynamic model we constructed for the proposed UAV. In Section 4, controller design architecture is introduced. In Section 5, the aerodynamic model of the unmanned aerial vehicle obtained using CFD method and the aerodynamic performance of different propellers of the aircraft are presented, and full section simulation experiments are conducted based on this. Finally, the sixth section concludes.

## 2. System Overview

This section introduces the overall design of the aircraft, full profile flight action process, and the design of the tilting mechanism.

### 2.1. Aircraft Design

The aircraft adopts 6 sets of electric propulsion units, of which four sets of propellers are installed at the end of the extended rack in the middle of the main wing, and the other two sets are installed at the end of the canard wing. The aircraft adopts an aerodynamic layout consisting of a fuselage, main wings, canard tails, and two symmetrically arranged external hangers. The specific layout and the motor numbers are shown in Figure 1.

A modified Liebeck LA5055 airfoil is employed for the main wing and the canard. The original Liebeck LA5055 airfoil [19] is designed to produce high lift at subsonic airspeed, and flow separation is delayed to decrease viscous drag. The airfoil is later modified to have the trailing edge slightly deflected upward to obtain a smaller pitch moment [20,21]. The installation angle is 0 deg and 2 deg for the main wing and the canard. A slightly higher installation angle for the canard results in an earlier stall than the main wing, thus, a deep stall is unlikely to happen during the flight. The canard will stall first, preventing the aircraft from pitching up further. Winglet was adopted for the main wing for lower induced drag and less intensive wingtip vortices. A pair of downward-facing V-tail is employed for directional stability and ground support and installed at the rear of the fuselage. NACA0012 airfoil profile is used for V-tail, and the angle between the two tail wings is 120 deg.

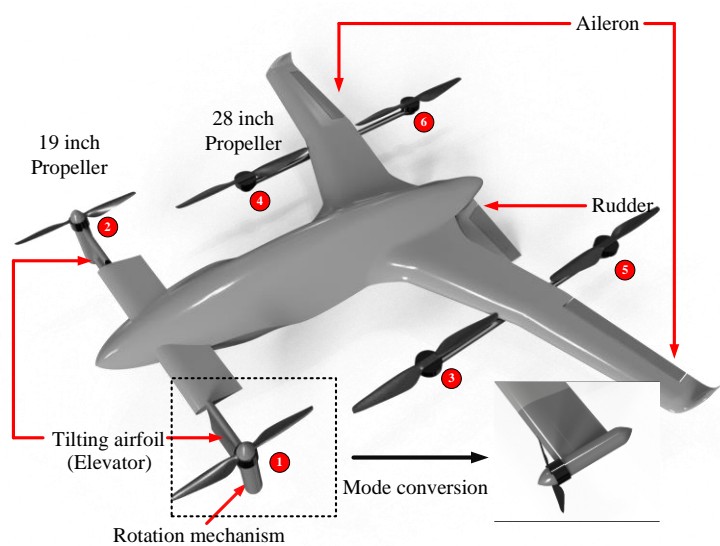

**Figure 1.** Overview of Target Aircraft Design.

To reduce the induced drag and structural mass caused by the canard wing located below the rotor during the vertical and transitional processes, a separate canard wing tilting section is designed here. This section can rotate with the rotor and serve as a separate deflection for the elevator to control the aircraft's attitude in a fixed wing mode. In fixed wing mode, the tilted portion of the canard can be used as an elevator to control the pitch stability of the fuselage. In terms of propeller configuration, the four propellers on the main wing are configured as propellers with high hovering efficiency, and the two propellers on the canard are configured as propellers with high cruising efficiency. Target aircraft design overall parameters are shown in Table 1.

**Table 1.** Target aircraft design overall parameters.

| Parameter | Description | Value |
|:---:|:---:|:---:|
| $m$ | Mass | 31.2 kg |
| $b$ | Wingspan | 2.7265 m |
| $\rho$ | Air density | 1.2250 kg/m$^3$ |
| $\bar{c}$ | Mean chord | 0.280922 m |
| $S$ | Wing surface area | 0.783078 m$^2$ |
| $V_{cruise}$ | Design cruise velocity | 29.6 m/s |

In the process of transitioning from a vertical to a horizontal flight orientation, the two tilt rotors on the canard wing propel the aircraft horizontally. After the aircraft reaches cruising speed, the four tilt rotors on the main wing stop running and adjust the blades to be parallel to the fuselage. At this time, only the two propellers on the canard provide cruise thrust. The operating logic during the backward transition is opposite to the forward transition. The mission profile of the compound tilt-rotor aircraft is shown in Figure 2.

It can be seen from the above aircraft design and transition method that compared with the standard compound VTOL, the target aircraft uses the front tilting mechanism to improve the power system utilization efficiency of the VTOL when hovering; Due to the presence of four fixed rotors at the rear, the forward tilt motor can still achieve force and torque balance in the aircraft's pitch dimension at a lower tilt angle during low-speed flight, expanding the tilt corridor compared to traditional tilt twin rotors; From the perspective of redundant control, the use of multiple actuators in a distributed hexacopter in rotor mode can effectively reduce the probability that a single actuator failure will cause the entire system to fail.

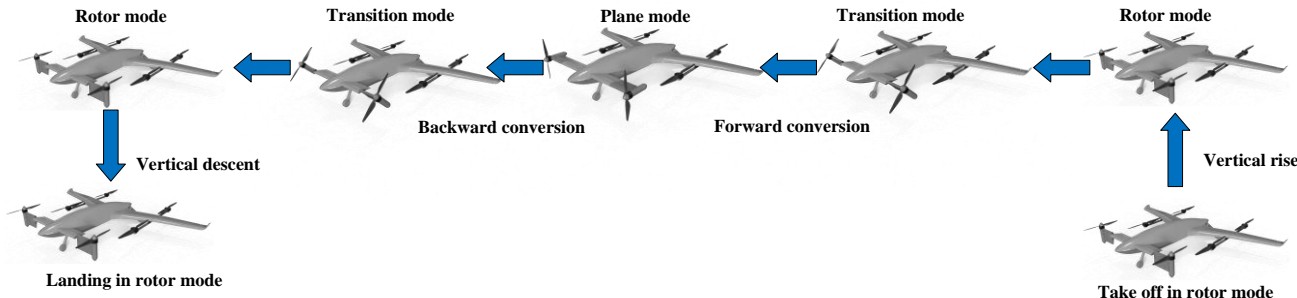

**Figure 2.** Mission profile.

## 2.2. Tilting Mechanism Design

The tilting mechanism and its driving device need to be designed separately. Here, we use a servo motor as the driver. The transmission mechanism consists of bevel gears and cylindrical gears, driven by a tilting motor for rotation, as shown in Figure 3. The bevel gear mainly changes the direction of the motor output shaft, and the tilting motor can be installed at the longitudinal end of the tilting mechanism, which is easy to be wrapped by the power short compartment skin, forming a slender cylindrical shape, and reducing aerodynamic resistance.

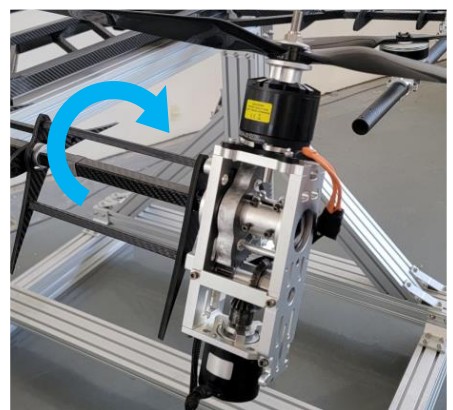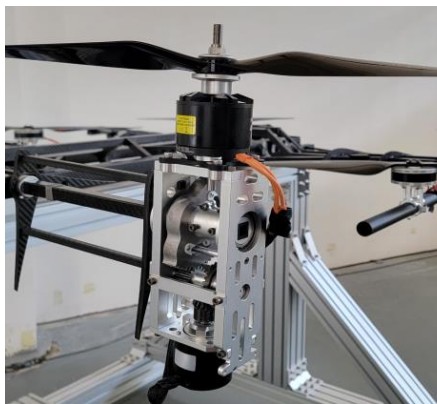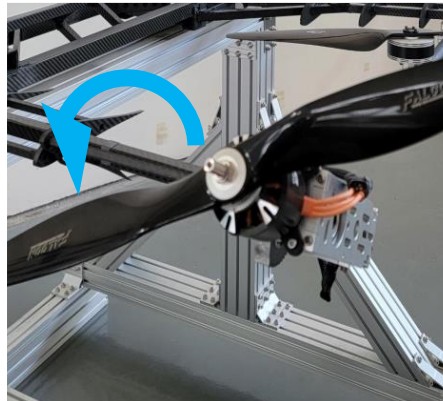

**Figure 3.** Tilting rotor drive mechanism and drive process.

The aluminum torque square tube in the wing tilting mechanism is used for wing load transfer, and the duck wing main beam is directly inserted into the inner diameter of the torque tube to fix the tilting mechanism and the fuselage. The tilting mechanism can achieve tilting within the range of $-45$ deg to 135 deg and achieve a 1:10 reduction ratio. The overall mass of the tilting unit is 1.1 kg, and the maximum driving angular velocity is 1.57 rad/s.

## 3. Dynamic Modeling

This section conducts flight dynamics modeling of the target tilt-rotor UAV. It briefly introduces the coordinate systems used in the modeling process and the mutual conversion between coordinate systems. Based on the concept of component modeling, dynamic models for the rotor, tilting/fixed wing, and fuselage were individually developed.

### 3.1. Definition

The modeling of flight dynamics and the design of the flight control system are inseparable from the coordinate system. In order to facilitate reference and calculation of the forces between various components, the coordinate systems mainly used in this article mainly include: inertial frame $\mathcal{F}^E = \{O_e, X_e, Y_e, Z_e\}$, body-fixed frame $\mathcal{F}^B = \{O_b, X_b, Y_b, Z_b\}$,

wind frame $\mathcal{F}^W = \{O_w, X_w, Y_w, Z_w\}$, as shown in Figure 4. The specific definition of the coordinate system can be found in [17].

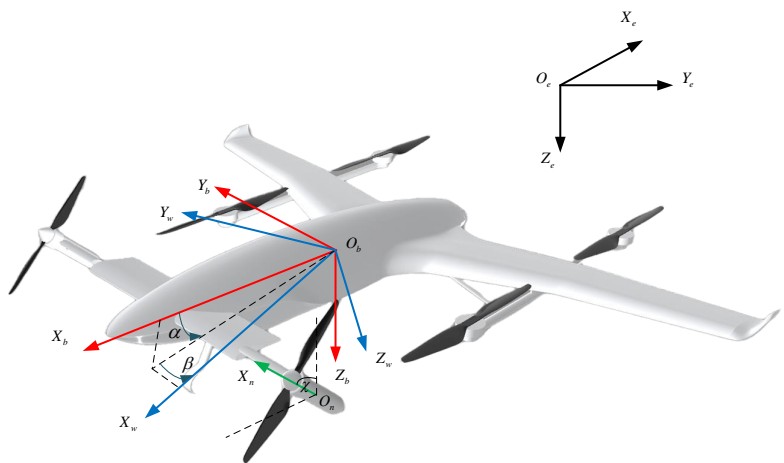

**Figure 4.** Coordinate System Definition.

In addition, we define the tilt angle of the tilt mechanism as $\chi$, where angle $\chi$ represents the tilt mechanism angle between the thrust vector $X_n$ and $-Z_b$. $X_n$ points towards the thrust direction of the tilting mechanism. According to this definition, the tilt angle of the tilt rotor aircraft in multirotor mode is 90 deg, and the tilt angle in fixed-wing mode is 0 deg. The tilt angle is within the range of $[-45 \deg, 135 \deg]$.

### 3.2. Modeling of Propeller Force and Torque

The thrust vector generated by each propeller on the canard ($i$ = 1, 2) in the body coordinate system is:

$$T^B_{rotor,i} = T_{rotor,i} R_{\chi i} e_3 \tag{1}$$

where $e_3 = [0, 0, 1]^T$ and the matrix expression of $R_{\chi i}$ is

$$R_{\chi i} = \begin{bmatrix} \cos \chi_i & 0 & -\sin \chi_i \\ 0 & 1 & 0 \\ \sin \chi_i & 0 & \cos \chi_i \end{bmatrix} \tag{2}$$

The incoming flow velocity of the propeller can be approximated as:

$$v_{ai} = R_{\chi i} v_b = [v_{axi}, v_{ayi}, v_{azi}] \tag{3}$$

where the airspeed vector $v_b = R^B_E (v - w)$, $v$ and $w$ represent the velocity of the tilt-rotor UAV and wind speed in the inertial frame respectively.

The values of thrust $T_i$ and torque $\tau_i$ can be estimated or calculated as follows:

$$\begin{aligned} T_{rotor,i} &= c_F(v, \omega)\omega_i^2 \\ \tau_i &= (-1)^{d_i} c_K(v, \omega)\omega_i^2 \end{aligned} \tag{4}$$

where the variables $c_F(v, \omega)$ and $c_K(v, \omega)$ represent the thrust and torque coefficients of the rotor, which are related to $v_{ax}$ and angular velocity $\omega$, coefficients of 1~2 rotors are defined as $c_{F1}$ and $c_{K1}$, coefficients of 3~6 rotors are defined as $c_{F2}$ and $c_{K2}$, The variable $d_i$ denotes the rotational direction of the $i$-th rotor along its axis.

When the incoming flow is not at a right angle to the propeller disk, the propeller will generate supplementary forces that are perpendicular to its rotational axis, as shown in Figure 5.

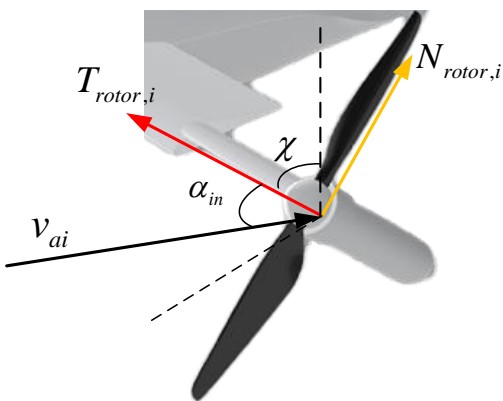

**Figure 5.** Force analysis of tilting propeller.

We use de-Young's empirical equation to estimate these normal forces [22], which are calculated as follows.

$$N_{rotor,i} = \frac{4.25\sigma_{\mathrm{e}} \sin(\beta_p + 8^\circ) f_i q_{\perp i} A_{\mathrm{disk}}}{1 + 2\sigma_{\mathrm{e}}} \tan \alpha_{in,i} \tag{5}$$

where $\beta_p$ represents blade pitch angle at 3/4 propeller radius, $q_\perp$ refers to the dynamic pressure determined by the free flow velocity component that is perpendicular to the propeller disk, $A_{\mathrm{disk}}$ represents the area of propeller, the angle at which the propeller is incident is denoted by $\alpha_{in}$, defined as $\arctan(v_{axi})/(v_{azi})$. According to [7], the effective solidity $\sigma_{\mathrm{e}}$ is

$$\sigma_{\mathrm{e}} = \frac{2Bc_{\mathrm{b}}}{3\pi R} \tag{6}$$

In the Equation (6), $B$ represents the quantity of blades on each propeller, $c_b$ stands for the average chord length of the blades, the variable $R$ denotes the radius of the tilting propeller, the thrust factor $f$ is denoted as [22]:

$$f = 1 + \frac{\sqrt{1 + T_c} - 1}{2} + \frac{T_c}{4(2 + T_c)} \tag{7}$$

where $T_c$ is defined (8) as $T_{\mathrm{c}} = T_{rotor}/(q_\perp A_{\mathrm{disk}})$, when $i = 1, 2$, the thrust vector $F^B_{rotor,i}$ and torque $M^B_{rotor,i}$ generated by each propeller, represented in body coordinate system as:

$$F^B_{rotor,i} = F_{rotor,i} R_{\chi i} \tag{8}$$

$$M^B_{rotor,i} = \tau_i R_{\chi i} e_3 = \tau_i [\sin \chi_i, 0, -\cos \chi_i]^T \tag{9}$$

where $F_{rotor,i} = [T_{rotor,i}, 0, N_{rotor,i}]$, for the majority of propeller-powered aircraft, the angle of attack of the propeller typically falls within a small value. During the transition period for tilt-rotor aircraft, the incident angle typically undergoes significant changes. Nevertheless, the low flight speed at this juncture results in a relatively small normal force produced by the propeller when considering the force and mass of the whole tiltrotor aircraft.

Afterwards, the thrust vector torque was modeled, and from the physical appearance of the aircraft, it can be concluded that for the front propellers 1 and 2, due to their own rotation:

$$d^B_{ri} = d_{fi} + R_{\chi i} d_{ei} \tag{10}$$

where $d_{f1} = [x_0, -l_0, -h_0]^T$, $d_{f2} = [x_0, l_0, -h_0]^T$, $d_{e1} = d_{e2} = [h_1, 0, 0]^T$, as shown in Figure 6.

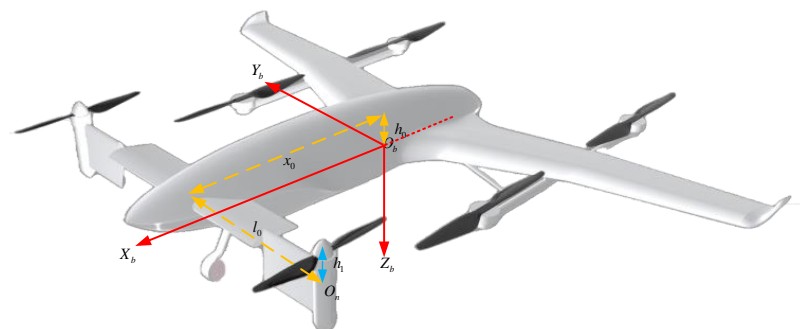

**Figure 6.** Definition of distance between tilting propeller and center of gravity.

For motors 3 to 6, due to their fixed connection with the body coordinates, $d_{ri}^B$ remains unchanged.

$$M_{thrust} = \sum_{i=1}^{6} T_i \times d_{ri}^B \tag{11}$$

The reaction torque generated by the rotor in free air due to rotor resistance is given by the following equation:

$$M_{rotate} = M_{rotor,1}^B + M_{rotor,2}^B + \begin{bmatrix} 0 \\ 0 \\ \tau_3 + \tau_4 + \tau_5 + \tau_6 \end{bmatrix} \tag{12}$$

*3.3. Aerodynamic Modeling and Simulation*

3.3.1. Modeling of Fixed Parts

The aerodynamic analysis of the main fixed part of the tilt-rotor UAV is similar to conventional aircraft. The airspeed $v_b$, attack angle $\alpha$, and sideslip angle $\beta$ are defined as

$$\begin{aligned} v_b &= \sqrt{v_{bx}^2 + v_{by}^2 + v_{bz}^2} \\ \alpha &= \tan^{-1}\left(\frac{v_{bz}}{v_{bx}}\right) \\ \beta &= \sin^{-1}\left(\frac{v_{by}}{v_b}\right) \end{aligned} \tag{13}$$

$F_m$ and $M_m$ generated by fixed parts (the fuselage and fixed wings) are expressed in the body frame system. The aerodynamic force and moment in the longitudinal direction is written as

$$F_m = R_W^B \begin{bmatrix} D_m \\ Y_m \\ L_m \end{bmatrix} \tag{14}$$

Among them, $D_m$, $Y_m$ and $L_m$ are respectively the drag, lift and lateral force expressed in the wind coordinate system. They can be written as

$$\begin{aligned} D_m &= \bar{q}SC_D(\alpha, \beta) \\ Y_m &= \bar{q}SC_Y(\alpha, \beta) \\ L_m &= \bar{q}SC_L(\alpha, \beta) \end{aligned} \tag{15}$$

where $S$ represents the reference area of the fixed part, dynamic pressure $\bar{q} = 0.5\rho v_b^2$. $C_L$, $C_D$ and $C_Y$ are the lift coefficient, drag coefficient and sideslip coefficient respectively. Furthermore, the aerodynamic moment acting on the fixed part can be represented as

$$M_m = \begin{bmatrix} b(C_l + C_{La}\delta_a)\bar{q}S \\ \bar{c}(C_m + C_{Me}\delta_e)\bar{q}S \\ b(C_n + C_{Nr}\delta_r)\bar{q}S \end{bmatrix} \tag{16}$$

In the Equation (16), $\bar{c}$ represents the average aerodynamic chord, while $b$ stands for the wingspan. The coefficients for rolling, pitching, and yawing are represented by $C_l$, $C_m$, and $C_n$ respectively. The effectiveness of the aileron, elevator, and rudder are represented by $C_{La}$, $C_{Me}$, and $C_{Nr}$ respectively. These coefficients are typically derived from data obtained through wind tunnel testing or CFD simulations.

### 3.3.2. Modeling of Tilting Section Canard

Accurately simulating the interaction between partially tilting canards and propeller-induced airflow is a challenging task. To briefly consider this interaction we turn to momentum theory [7]. The induced velocity $v_p$ on the propeller is given as:

$$v_p = \left( -\frac{v_{ax}}{2} + \sqrt{\frac{v_{ax}^2}{4} + \frac{T_{rotor}}{2\rho A_{\text{disk}}}} \right) \tag{17}$$

From this it can be concluded that:

$$v_{rt} = \left[ (v_p + v_{ax})^2 + (v_{ay})^2 + (v_{az})^2 \right]^{1/2}, \alpha_f = \arctan(v_{az}) / (v_p + v_{ax}) \tag{18}$$

where $v_{rt}$ is the resultant speed at the tilting canard, and $\alpha_f$ is the angle of attack of the tilting section canard. Therefore, the lift $L_{fi}$ and drag $D_{fi}$ ($i = 1, 2$) on the tilting canard are written as

$$\begin{aligned} L_{fi} &= 0.5\rho S_{fi} C_{L_f} v_{rt}^2 \\ D_{fi} &= 0.5\rho S_{fi} C_{D_f} v_{rt}^2 \end{aligned} \tag{19}$$

where $C_{L_f} = C_{L_{f0}} + C_f \alpha_f, C_{D_f} = C_{D_{f0}} + C_{L_f}^2 / \left( \pi A_f e_f \right), S_{fi}$ is the area of the tilting canard. $C_{L_{f0}}$ is the lift coefficient at 0 angle of attack. $C_f$ is the rate of change of lift coefficient with angle of attack $\alpha_f$ to the tilting canard, and $C_{D_{f0}}$ is the drag coefficient at 0 angle of attack. According to [6], $e_f$ is defined as the Oswald efficiency factor and empirical expression is $e_f = 1.78 \left( 1 - 0.045 A_f^{0.68} \right) - 0.46$, $A_f$ is the aspect ratio of the tilting canard.

In the process of transitioning from a vertical to a horizontal flight orientation, the tilting portion of the canard needs to consider separation flow conditions. We need a wing aerodynamic model that can also predict lift and drag outside the linear lift region. We use a finite-length rectangular wing model developed by Tangler and Ostowari [23] based on experimental data and Viterna and Corrigan's model [24]. The lift and drag coefficients after stall are given by:

$$C_{Lf} = A_1 \sin 2\alpha_f + A_2 \frac{\cos^2 \alpha_f}{\sin \alpha_f}, \tag{20}$$

$$C_{Df} = B_1 \sin \alpha_f + B_2 \cos \alpha_f \tag{21}$$

where

$$\begin{aligned} A_1 &= \frac{C_1}{2} & A_2 &= (C_{L_s} - C_1 \sin \alpha_s \cos \alpha_s) \frac{\sin \alpha_s}{\cos^2 \alpha_s} & C_1 &= 1.1 + 0.018 A_f \\ B_1 &= C_{D_{\max}} & B_2 &= \frac{C_{D_s} - C_{D_{\max}} \sin \alpha_s}{\cos \alpha_s} & C_{D_{\max}} &= \frac{1.0 + 0.065 A_f}{0.9 + t/c} \end{aligned} \tag{22}$$

where $\alpha_s$ represents the angle of attack at which stall occurs, while $C_{L_s}$ and $C_{D_s}$ denote the lift coefficient and drag coefficient corresponding to stall respectively. The tilting canard's thickness-to-chord ratio is represented by $t/c$.

The moment generated by the canard tilt is $M_f = \sum_{i=1}^{2} R_{\chi_i}^T F_{fi}^N \times d_{fi}^B$, where $F_{fi}^N = [L_{fi}, 0, D_{fi}]^T$, The aerodynamic force $\boldsymbol{F_{aero}}$ and moment $\boldsymbol{M_{aero}}$ are expressed in the body coordinate system:

$$\begin{aligned} \boldsymbol{F_{aero}} &= \boldsymbol{F_m} + \sum_{i=1}^{2} R_{\chi_i}^T F_{fi}^N \\ \boldsymbol{M_{aero}} &= \boldsymbol{M_m} + \boldsymbol{M_f} \end{aligned} \tag{23}$$

*3.4. Total Force and Moment Modeling*

The total force can be expressed through the subsequent mathematical equation:

$$\boldsymbol{F} = \boldsymbol{F_{aero}} + \boldsymbol{F_g} + \boldsymbol{F_{rotor}} \tag{24}$$

where $\boldsymbol{F_{aero}}$ is the aerodynamic force in the body frame system, the variable $\boldsymbol{F_g}$ represents the gravitational force within the body frame system, and $\boldsymbol{F_{rotor}}$ is the rotor pull force in the body frame system. The specific expressions are:

$$\boldsymbol{F_{rotor}} = \boldsymbol{F^B_{rotor,1}} + \boldsymbol{F^B_{rotor,2}} + \sum_{i=3}^{6} \begin{bmatrix} 0 \\ 0 \\ -1 \end{bmatrix} T_i \tag{25}$$

The total external moment in the body frame system can be expressed as:

$$\boldsymbol{M} = \boldsymbol{M_{rotate}} + \boldsymbol{M_{thrust}} + \boldsymbol{M_{aero}} \tag{26}$$

## 4. Flight Control System Design

The diagram illustrating the structure of the flight control system can be observed in Figure 7. The system comprises four levels: the position controller, the vertical takeoff and landing fusion device, the ADRC attitude controller, and the mixer. At the same time, the entire unmanned aerial vehicle control system adopts a hierarchical control architecture.

The position controller processes the specified position data to determine the desired attitude and propulsion, which is subsequently transmitted to the VTOL fusion system. Following this, the VTOL fusion module conveys all anticipated attitude from position controllers to the attitude controller. In addition, the VTOL fusion device also sends the expected thrust to the mixer for further processing.

The attitude controller converts the desired attitude into control torque and sends it to the mixer. The mixer uses an appropriate torque distribution scheme based on the current flight mode of the aircraft to map torque and thrust to the actuator for aircraft motion control. Below is a detailed introduction to several modules:

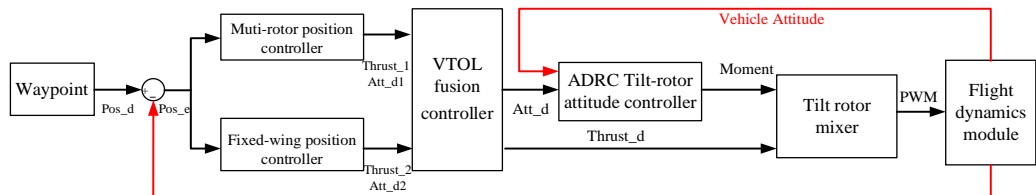

**Figure 7.** Design block diagram of hierarchical control system.

*4.1. Position Controller*

For the convenience of subsequent flight control system software porting and flight testing, our position controller architecture mainly refers to the existing framework of PX4 autonomous flight software v1.13 [25]. The multirotor position controller comes from [26], and the fixed wing position controller mainly comes from [27]. Our system directly adopts these two mature modules.

*4.2. Vertical Takeoff and Landing Fusion Controller*

The vertical takeoff and landing fusion controller plays a crucial role in the overall control architecture by integrating the attitude and thrust set points, along with the pitch angle, from the multirotor and fixed-wing position controllers. The output of the VTOL

fusion controller is the expected thrust $T_{sp}$ in the $x$ and $z$ directions and a desired attitude set point $\psi_{sp}$, where $T_{sp,x}$ and $T_{sp,z}$ can be calculated empirically as:

$$T_{sp,x} = T_{fc} constrain(\frac{|v_b|-15}{V_c-15}, 0, 1)$$
$$T_{sp,z} = T_{mc}(1 - constrain(\frac{|v_b|-15}{V_c-15}, 0, 1)) \tag{27}$$

In Equation (27), the $constrain(a, b, c)$ represents that $a$ needs to satisfy its own numerical value within the range of $b \sim c$. $T_{mc}$ and $T_{fc}$ represent the expected thrust set points for the position control outputs of the multirotor and fixed-wing, respectively. The parameter $V_c$ represents the minimum threshold velocity that the aircraft needs to surpass in order to transition into full fixed wing mode.

Similarly, the attitude set point $\psi_{sp}$ is linearly integrated through multirotor controllers and fixed wing controllers, and its expression is

$$\psi_{sp} = \alpha\psi_{mc} + (1 - \alpha)\psi_{fw} \tag{28}$$

$$\alpha = constrain(1 - \frac{|v_b|}{V_c}, 0, 1) \tag{29}$$

where $\psi_{mc}$ and $\psi_{fw}$ represent the expected attitude set points for the position control outputs of the multirotor and fixed wing, respectively.

The tilt controller obtains the forward airspeed $v_b$ and outputs it to the tilt angle $\chi$. Based on experience [28], it is found that the airspeed to the tilt angle has a linear mapping:

$$\chi = \frac{\pi}{2} - constrain(k_\chi(|v_b| - V_\chi), 0, \frac{\pi}{2}) \tag{30}$$

where $V_\chi$ is the threshold velocity for starting tilting, $k_\chi$ is the coefficient of tilt direction.

If the forward airspeed is less than $V_\chi$, the propeller points upward and the tilt mechanism remains unchanged. As the forward velocity increases, propellers No. 1 and 2 progressively incline forward until the aircraft transitions into a fixed-wing arrangement at $V_c$.

### 4.3. Attitude Controller

The function of the attitude controller is to convert the desired control attitude set point output by the upper-level VTOL fusion controller into a desired torque $\tau$ and output it to the lower-level mixer, so that the aircraft can accurately track the attitude set point. Here we use the ADRC control unit instead of the traditional PID controller. ADRC technology does not rely on models, can handle various internal uncertainties, and is highly robust. According to reference [29], basic ADRC consists of three parts: Differential tracker (TD), Extended state observer (ESO) and Nonlinear law of state error feedback (NLSEF). The diagram illustrating the configuration of ADRC attitude controller is presented in Figure 8.

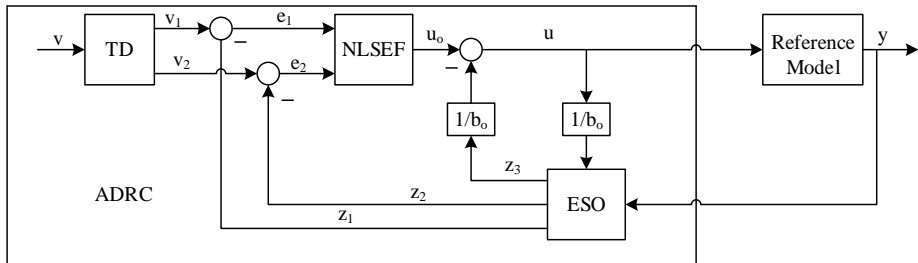

**Figure 8.** The structure of the ADRC controller [29].

### 4.3.1. Tracking Differentiator

The function of TD is to arrange signal transitions and provide relatively smooth control signals, with the aim of resolving the contradiction between response speed and overshoot, its expression is:

$$\begin{cases} r_1(k+1) = r_1(k) + hr_2(k) \\ r_2(k+1) = r_2(k) + hfst(r_1(k) - v(k), r_2(k), \delta, h) \end{cases} \tag{31}$$

where $r_1(k)$ represents the tracking signal of $v(k)$, $r_2(k)$ represents the differential signal of $v(k)$, The parameter $\delta$ is responsible for determining the velocity of tracking, and $h$ is the step size. According to reference [11], the expression of $fst(x_1, x_2, \delta, h)$ is:

$$fst(x_1, x_2, \delta, h) = - \begin{cases} \delta \text{sign}(a) & |a| > d \\ \delta \frac{a}{d} & |a| \le d \end{cases} \tag{32}$$

$$a = \begin{cases} x_2 + \frac{(a_0 - d)}{2} \text{sign}(y) & |y| > d_0 \\ x_2 + \frac{y}{h} & |y| \le d_0 \end{cases} \tag{33}$$

$$\begin{cases} d = \delta h \\ d_0 = hd \\ y = x_1 + hx_2 \\ a_0 = \sqrt{d^2 + 8\delta|y|} \end{cases} \tag{34}$$

### 4.3.2. Extended State Observer

The Extended State Observer has the capability to estimate both internal disturbances within the system and external disturbances affecting the system. The design and functionality of the ESO can be characterized by its structure as follows:

$$\begin{cases} e = z_1 - y \\ \dot{z}_1 = z_2 - \beta_1 e \\ \dot{z}_2 = z_3 - \beta_2 fal(e, \alpha_1, \delta) + bu \\ \dot{z}_3 = -\beta_3 fal(e, \alpha_2, \delta) \end{cases} \tag{35}$$

where $\beta_i > 0 (i = 1, 2), \alpha_1 = 0.5, \alpha_2 = 0.25$. The function of the saturation function $fal(e, \alpha, \delta)$ is to suppress signal jitter [29].

$$fal(e, \alpha, \delta) = \begin{cases} \frac{e}{\delta^{1-\alpha}} & |e| \le \delta \\ |e|_\alpha \text{sgn}(e) & |e| > \delta \end{cases} \tag{36}$$

### 4.3.3. Nonlinear Law of State Error Feedback

The NLSEF is a control law that integrates the discrepancy between the output of the tracking differentiator and the system state in a nonlinear manner. The expression of the NLSEF control law is outlined as follows:

$$\begin{cases} e_1 = v_1 - z_1 \\ e_2 = v_2 - z_2 \\ u = \beta_1 fal(e_1, \alpha_1, \delta) + \beta_2 fal(e_2, \alpha_2, \delta) \end{cases} \tag{37}$$

where $0 < \alpha_1 < 1 < \alpha_2, k_p = \beta_1, k_d = \beta_2$.

### 4.4. Control Distribution

The objective of the control allocation unit is to assign the anticipated thrust and moments in roll, pitch, and yaw along the $x$ and $z$ axes of the fuselage to the 14 actuators present on the target aircraft. In order to streamline computational processes, we have

opted to simplify the system by reducing the number of actuators from 14 to 11: two elevators have the same tilt angle in the same direction, two ailerons have the same tilt angle in the opposite direction, and the rudder has the same tilt angle in the same direction.

Therefore, the goal of control assignment is to determine the state vector $N = [f_1 \sim f_6, \chi_L, \chi_R, \delta_a, \delta_e, \delta_r]$ of the actuator given $U_M = [T_{x,des}, T_{z,des}, L_{des}, M_{des}, N_{des}]$, where $f_i = c_{F1}\omega_i{}^2$, $\chi_L$ and $\chi_R$ are the inclination angles of the left and right tilting mechanisms, respectively.

Firstly, the rudder deviation angle can be derived from the following equation:

$$\begin{bmatrix} \delta_a \\ \delta_e \\ \delta_r \end{bmatrix} = \begin{bmatrix} 1 & 0 & 0 \\ 0 & 1 & 0 \\ 0 & 0 & 1 \end{bmatrix} \begin{bmatrix} \frac{2(1-J(v))M_{Tx}}{\rho v_b^2 S b C_{La}} \\ \frac{2(1-J(v))M_{Ty}}{\rho v_b^2 S \bar{c} C_{Me}} \\ \frac{2(1-J(v))M_{Tz}}{\rho v_b^2 S b C_{Nr}} \end{bmatrix} \tag{38}$$

where $M_T = [M_{Tx}, M_{Ty}, M_{Tz}]^T$ is the moment vector, $J(v)$ is a function related to airspeed to weigh the moment contribution of the rudder surface in the pitch, roll and yaw directions, calculated as follows:

$$J(v) = 1 - constrain\left(\frac{|v_b| - 15}{V_c - 15}, 0, 1\right) \tag{39}$$

The thrust of the six rotors can be obtained by the following equation:

$$\begin{bmatrix} T_{x,des} \\ T_{z,des} \\ J(v)L_{des} \\ J(v)M_{des} \\ J(v)N_{des} \end{bmatrix} = \left( A \right) \begin{pmatrix} f_1 \\ f_2 \\ f_3 \\ f_4 \\ f_5 \\ f_6 \end{pmatrix} \tag{40}$$

where $A$ is

$$A = \begin{bmatrix} \sin(\chi_R) & \sin(\chi_L) & 0 & 0 & 0 & 0 \\ \cos(\chi_R) & \cos(\chi_L) & \frac{c_{F2}}{c_{F1}} & \frac{c_{F2}}{c_{F1}} & \frac{c_{F2}}{c_{F1}} & \frac{c_{F2}}{c_{F1}} \\ -L_1\sin\theta_1\cos\chi_R & L_1\sin\theta_1\cos\chi_L & -\frac{c_{F2}}{c_{F1}}L_2\sin\theta_2 & \frac{c_{F2}}{c_{F1}}L_2\sin\theta_2 & -\frac{c_{F2}}{c_{F1}}L_3\sin\theta_3 & \frac{c_{F2}}{c_{F1}}L_3\sin\theta_3 \\ -L_1\cos\theta_1\cos\chi_R & -L_1\cos\theta_1\cos\chi_L & -\frac{c_{F2}}{c_{F1}}L_2\cos\theta_2 & -\frac{c_{F2}}{c_{F1}}L_2\cos\theta_2 & \frac{c_{F2}}{c_{F1}}L_3\cos\theta_3 & \frac{c_{F2}}{c_{F1}}L_3\cos\theta_3 \\ \frac{c_{K1}}{c_{F1}}\cos\chi_R + L_1\sin\chi_R\sin\theta_1 & -\frac{c_{K1}}{c_{F1}}\cos\chi_L - L_1\sin\chi_L\sin\theta_1 & -\frac{c_{K2}}{c_{F1}} & \frac{c_{K2}}{c_{F1}} & \frac{c_{K2}}{c_{F1}} & -\frac{c_{K2}}{c_{F1}} \end{bmatrix}$$

where $L_1$, $L_2$ and $L_3$ respectively represent the vectors between the center of mass of the aircraft and the projection points of motors 1~2, 3~4, and 5~6 on the xoy plane in the body frame, $\theta_i$ represents the angle between $L_i$ and $O_b X_b$. The derivation of the mixed control matrix $A$ can be found in reference [30].

## 5. Simulation of Full Flight Mode

### 5.1. Simulation System

We modeled the dynamics of the target tiltrotor unmanned aerial vehicle flight process in Simulink based on the content of the Section 3 and simulated the velocity and position response of the tiltrotor control architecture (position controller, ADRC attitude controller, VTOL fusion and mixer) developed in the Section 4.

The complete simulation system is illustrated in Figure 9, featuring a highly modular model comprising four distinct subcomponents: aircraft dynamics, flight controller, control command module, and visualization module. The control command module has the capability to receive various types of set control commands, including dynamic position coordinate points and velocity vectors. The visualization module adopts the popular FlightGear software [31], which has a relatively complete interface with MATLAB Simulink, and its display is also more realistic.

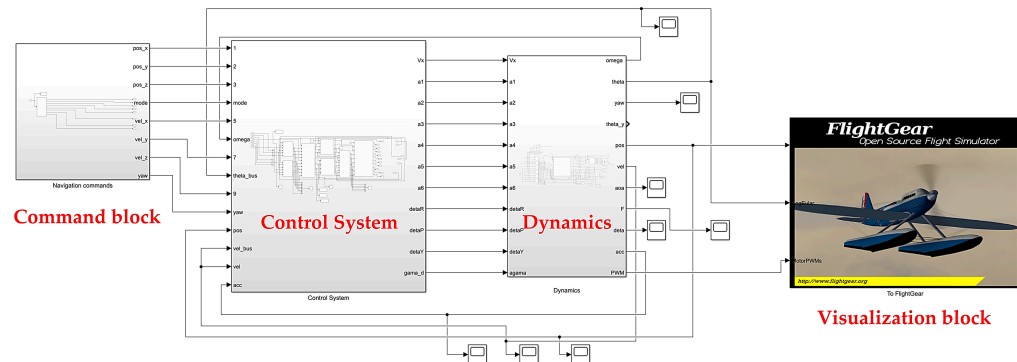

**Figure 9.** Tilt-rotor UAV simulation system in MATLAB Simulink.

In addition, the constant values used in the full mode flight simulation process are shown in Table 1. The propellers of the target tiltrotor aircraft all use standard APC airfoils, the specific $c_F$ and $c_K$ of propellers 1~6 can be obtained on the APC official website [32] and they will not be listed in the article.

Lift coefficient $C_L$, drag coefficient $C_D$, sliding coefficient $C_Y$, and three-axis torque coefficient of the target aircraft $C_m$, $C_n$, $C_l$ are shown in Figures 10 and 11, which were obtained from CFD simulations. The aileron coefficients $C_{La}$, the elevator coefficients $C_{Me}$ and the rudder coefficients $C_{Nr}$ is set to 0.2314, 0.5560 and 0.0581 respectively.

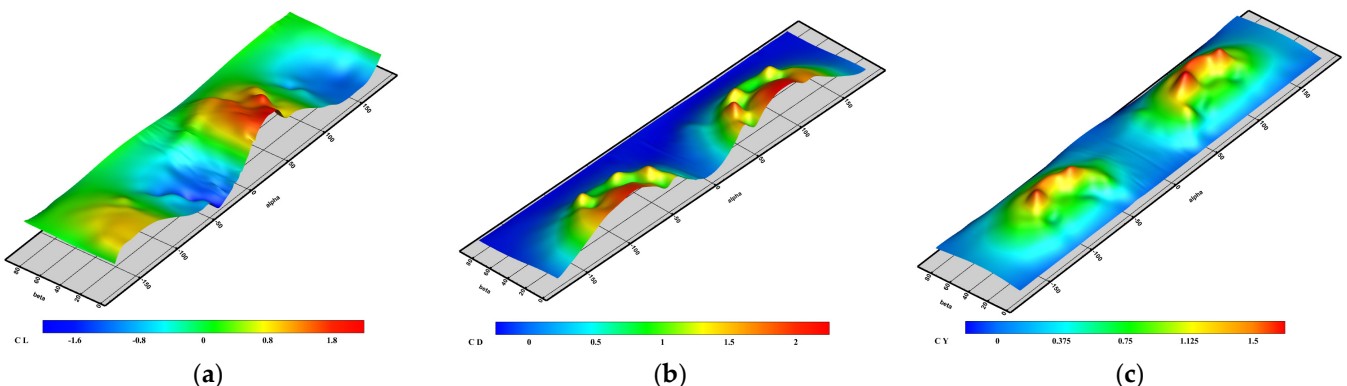

**Figure 10.** CFD simulation values for (**a**) lift coefficient $C_L$, (**b**) drag coefficient $C_D$, and (**c**) sideslip coefficient $C_Y$ of the fixed parts of the target tiltrotor.

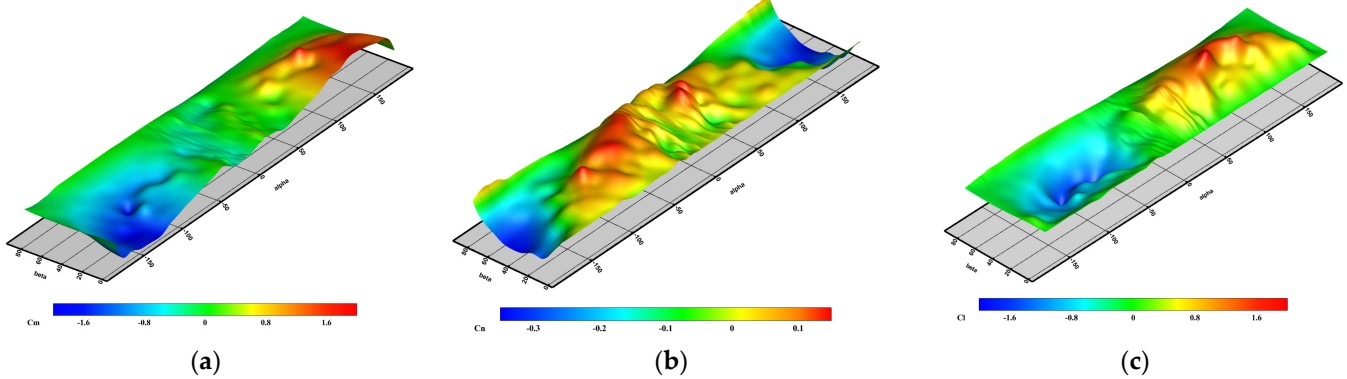

**Figure 11.** CFD simulation values for (**a**) pitch moment coefficient $C_m$, (**b**) sideslip moment coefficient $C_n$, and (**c**) rolling moment coefficient $C_l$ of the fixed parts of the target tiltrotor.

*5.2. Velocity Response Control Simulation*

The simulation of velocity response control in flight profile comprises five components: taking off in multirotor mode, accelerating flight in transition mode, cruising in aircraft

mode, decelerating flight, and landing in multirotor mode. The primary parameters of the control system pertaining to velocity response are outlined in Table 2, it is worth noting that the parameters of ADRC mainly based on [15].

**Table 2.** Main parameters of the control system in velocity response.

| Parameter | PID of Velocity Loop | Parameter | ADRC |
|---|---|---|---|
| $P_{mc}$ | (1.2 1.2 0.8) | r | (20 10 20) |
| $I_{mc}$ | (0.02 0.01 0.01) | h | (0.01 0.01 0.01) |
| $D_{mc}$ | (0.001 0.001 0.001) | $b_0$ | (30 30 40) |
| $P_{fw}$ | (0.25 0.25 0.15) | $\beta_{01}$ | (200 100 50) |
| $I_{fw}$ | (0.03 0.02 0.02) | $\beta_{02}$ | (10 5 5) |
| $D_{fw}$ | (0.001 0.001 0.001) | $\beta_1$ | (200 200 100) |
|  |  | $\beta_2$ | (1000 1000 500) |
|  |  | $\beta_3$ | (50 50 25) |

In order to evaluate the VTOL model and the suggested controller, various velocity commands were applied to the simulation model at different intervals. The flight process is as follows: within 0–15 s, the tiltrotor takes off vertically in multirotor mode to a certain altitude and hovers in multirotor mode; In 15–30 s, the velocity of forward flight progressively rises, and ultimately shifts into fixed wing mode; Cruise in fixed wing mode within 30–40 s; In 40–65 s, the forward velocity of the aircraft diminishes, leading to a gradual transition of the aircraft mode to multirotor mode; In 65–85 s, the UAV hovers first and then lands on the ground. To reduce the impact of rotor 1 and 2 on the rear propeller during the tilting process, we set $V_\chi$ to 16.5 m/s and $V_c$ to 25 m/s.

Initially, the simulation was conducted in the absence of any external disruptions. According to the data presented in Figure 12, there is an absence of notable variation in the pitch angle when operating in multirotor mode. When the transition begins, there is a sudden change of about 0.18 deg in the pitch direction. When the cruise stabilizes, there is a change of about −0.03 deg in the pitch direction. As the deceleration increases, the pitch direction angle gradually increases to 0.15 deg, and then the pitch direction rapidly decreases. The roll and yaw angles undergo significant changes during the overall flight process.

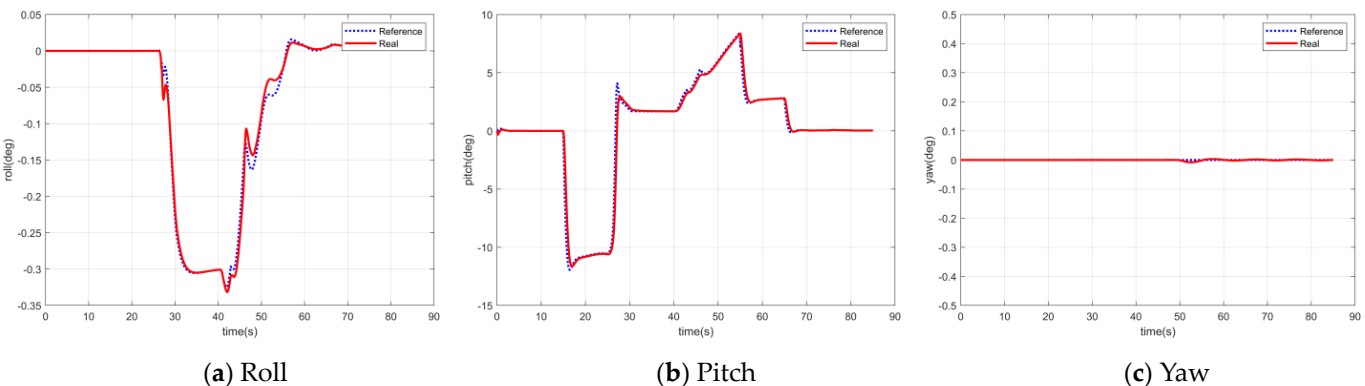

(**a**) Roll        (**b**) Pitch        (**c**) Yaw

**Figure 12.** Flight state response curve of attitude in velocity response control simulation without disturbance.

The response results under velocity mode are shown in Figure 13. The velocity tracking accuracy and response velocity of the three axes are good, and their steady-state errors are less than 0.05 m/s in both multirotor and fixed wing modes. The maximum error of the three-axis velocity tracking occurs during the transition from multirotor to fixed wing and from fixed wing to rotor. On the one hand, because the transition process of tiltrotor unmanned aerial vehicles is a variable structure process with strong coupling

characteristics, on the other hand, there are significant changes in the control variables between fixed wing and rotor controllers during the switching process.

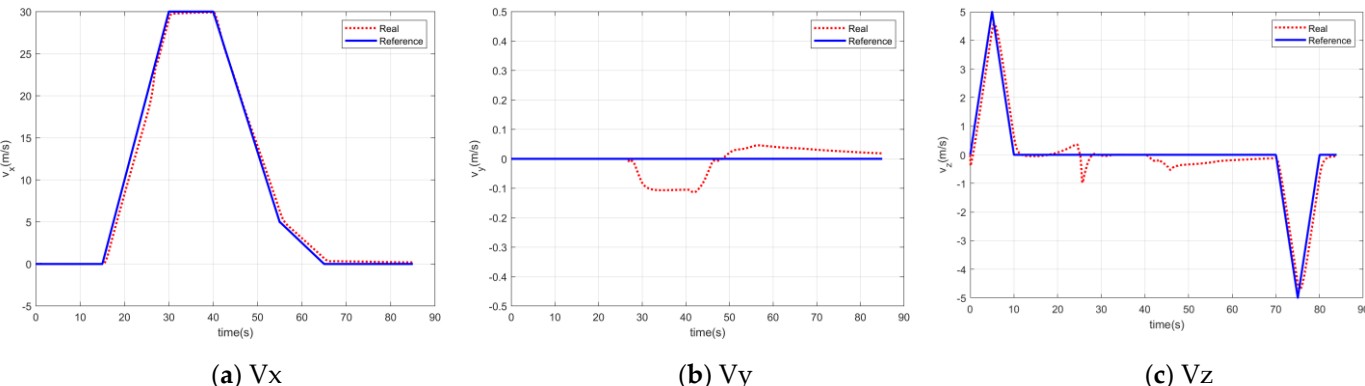

**(a)** Vx      **(b)** Vy      **(c)** Vz

**Figure 13.** Flight state response curve of velocity in velocity response control simulation without disturbance.

Figure 14 shows the required engine thrust in different modes. As the mode switches to fixed wing mode, the required 1st and 2nd speeds rapidly increase to reach the upper limit of the speed. At the same time, the speed of motors 3–6 decreased to 0. As the fixed wing mode changed to multirotor mode, the speeds of motors 1 and 2 gradually decreased. At the same time, the speed of motors 3–6 increased and finally stabilized within a fixed range. Although using the method in Section 4.2 for transition flight may not necessarily be the most fuel efficient, it is stable and reliable in practice, while also expanding the transition envelope of the drone to a certain extent.

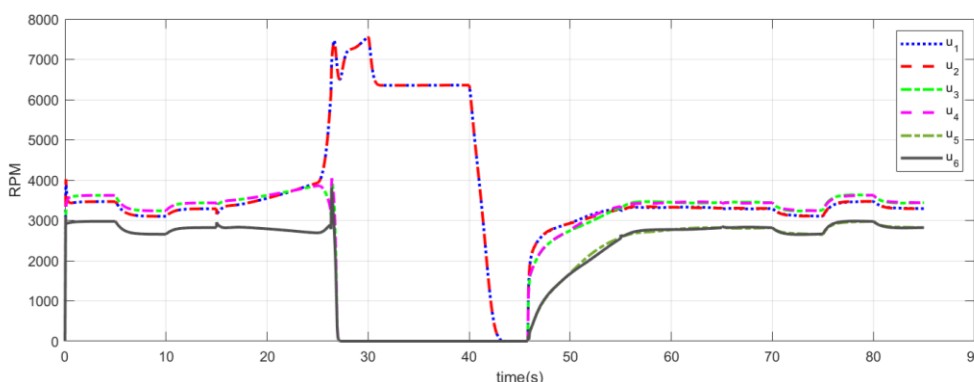

**Figure 14.** Flight state response curve of Rotors' RPM in velocity response control simulation without disturbance.

Due to the external disturbances experienced by the aircraft during flight, we add white noise with a sampling frequency of 0.1 s and an amplitude of 0.1 to the attitude measurement values to simulate the observation errors of IMU sensors; We also add white noise with a sampling frequency of 0.1 s and an amplitude of 0.05 to the measured velocity to simulate GPS measurement errors [33]. The remaining configurations align with the simulation that does not involve any disturbance.

From Figure 15, with the addition of disturbances, there is a random deviation of 0.02~0.15 deg between the roll, pitch and yaw channels and the reference value. This is due to the addition of external and internal disturbances to the system. The maximum error in the pitch direction is 1.1 deg. The three-axis angle can still stably track the desired attitude angle, indicating that the system has a certain level of disturbance resistance. The findings from the simulation demonstrate that the controller exhibits certain anti-

disturbance attributes and can stabilize the aircraft under certain disturbances and achieve smooth transition and flight in different modes.

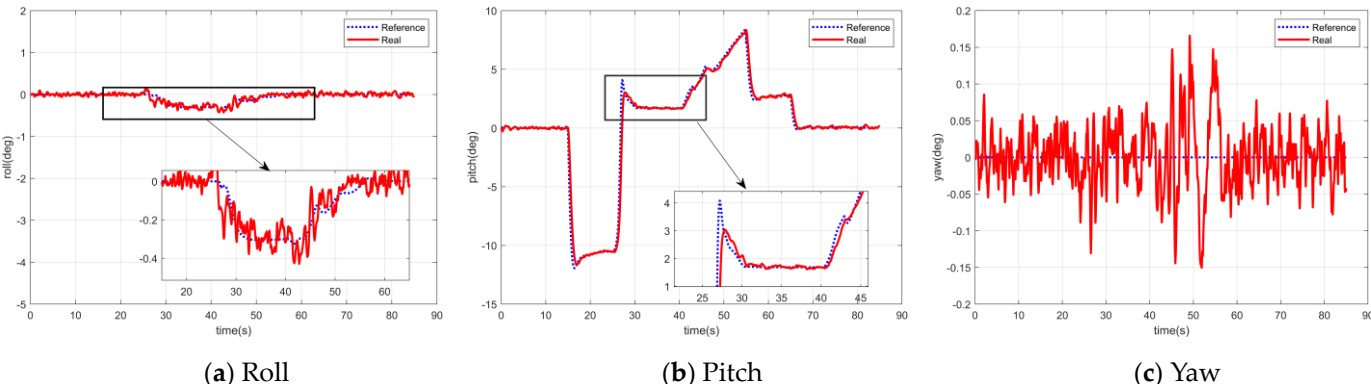

**Figure 15.** Flight state response curve of attitude in velocity response control simulation with disturbance.

*5.3. Position Mode Control Simulation*

Based on the velocity response tracking test, we conducted a full flight mode trajectory tracking response test, we adopt position mode, and the input to the controller is the expected position trajectory with time information.

To get closer to the real flight environment, we added white noise with amplitudes of 0.1, 0.05 and sampling frequencies of 0.1 s to the attitude, velocity measurements to simulate the observation error of IMU and GPS sensors. In addition, external wind disturbances can interfere with UAV flight. We introduced an external wind perturbation characterized by a wind velocity of $v_f = 0.5\sin(0.5t)$ to the three-dimensional inflow affecting the vehicle [34]. The primary variables of the control system are presented in Table 3. The flight command path is shown in Figure 16 (the simulation video in position mode can be found in: https://www.youtube.com/watch?v=pcORCEPILsA, accessed on 8 March 2024).

**Table 3.** Main parameters of position mode control test.

| Parameter | PID of Position Loop | Parameter | PID of Velocity Loop | Parameter | ADRC |
|---|---|---|---|---|---|
| $P_{mc}$ | (0.8 0.8 0.6) | $P_{mc}$ | (1.2 1.2 0.8) | r | (20 10 20) |
| $I_{mc}$ | (0 0 0) | $I_{mc}$ | (0.02 0.01 0.01) | h | (0.01 0.01 0.01) |
| $D_{mc}$ | (0.001 0.001 0.001) | $D_{mc}$ | (0.001 0.001 0.001) | $b_0$ | (30 30 40) |
| $P_{fw}$ | (0.7 0.8 0.4) | $P_{fw}$ | (0.25 0.25 0.15) | $\beta_{01}$ | (200 100 50) |
| $I_{fw}$ | (0 0 0) | $I_{fw}$ | (0.03 0.02 0.02) | $\beta_{02}$ | (10 5 5) |
| $D_{fw}$ | (0.001 0.001 0.001) | $D_{fw}$ | (0.001 0.001 0.001) | $\beta_1$ | (200 200 100) |
| | | | | $\beta_2$ | (1000 1000 500) |
| | | | | $\beta_3$ | (50 50 25) |

The response result of the angle is shown in Figure 17. Due to sensor measurement errors, there is a random deviation of 0.05~0.3 deg between the roll, pitch, yaw channels and the reference value. Additionally, due to external periodic disturbances, there are periodic fluctuations in the yaw direction in fixed wing mode, with a fluctuation range of within 0.5 deg. From the simulation results, the angles of the three axes in multirotor mode and fixed-wing modes can track the command angle well.

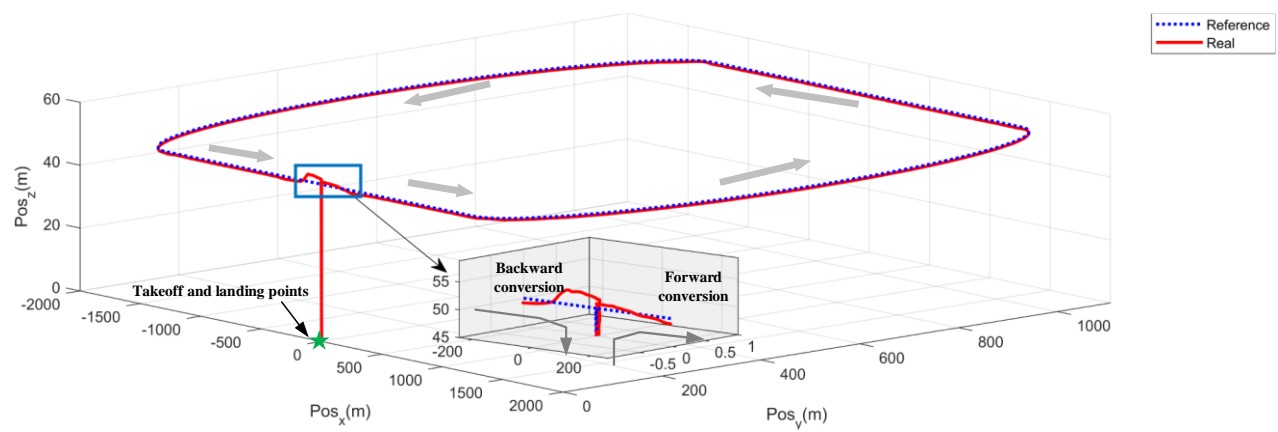

**Figure 16.** 3D trajectory tracking in UAV position mode.

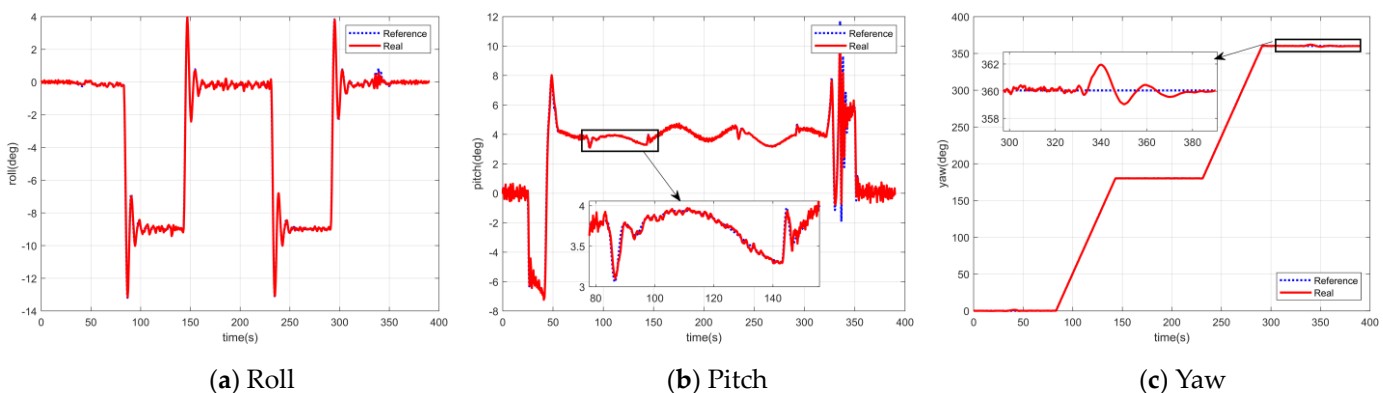

(**a**) Roll  (**b**) Pitch  (**c**) Yaw

**Figure 17.** Flight state response curve of attitude in position mode control simulation.

The velocity response result is shown in Figure 18. The velocity tracking accuracy and response velocity of the three-axis in position mode are both good. In multirotor and fixed-wing modes, there is a stochastic variation ranging from 0.05 to 0.1 m/s observed in X, Y, and Z axes. During the forward and backward transition of the Z-axis, small amplitude oscillations occur and then converge rapidly.

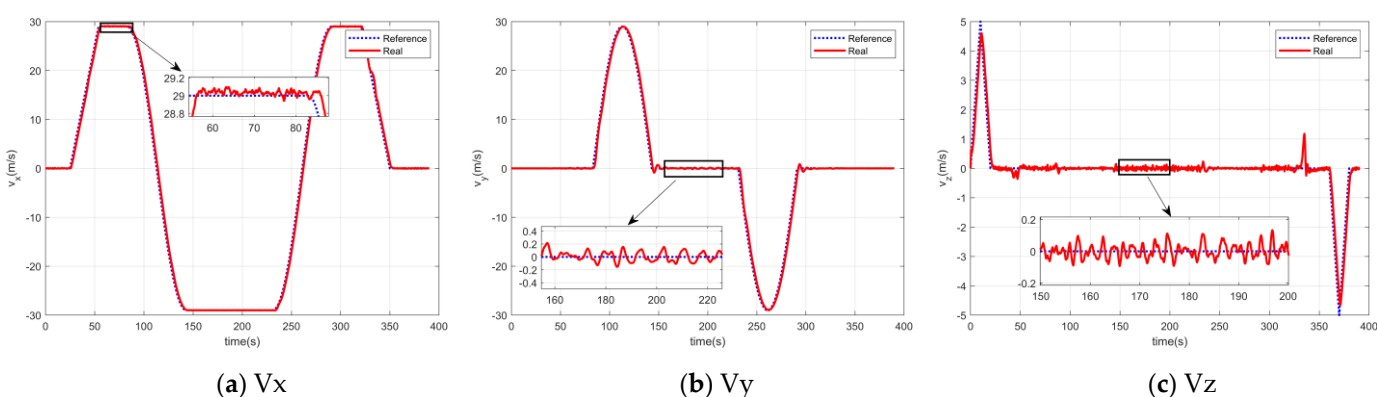

(**a**) Vx  (**b**) Vy  (**c**) Vz

**Figure 18.** Flight state response curve of velocity in position mode control simulation.

The outcomes of the positional simulation are depicted in Figure 19. These results indicate that the F-ADRC control architecture maintains high accuracy in the full mode flight trajectory tracking process of the tiltrotor. The maximum error in maintaining altitude is 1.6 m, which occurs at the beginning of the transition from fixed wing mode to multirotor

mode. In both horizontal cruise and multirotor modes, the average control error of position remains within 0.2 m despite external and internal disturbances.

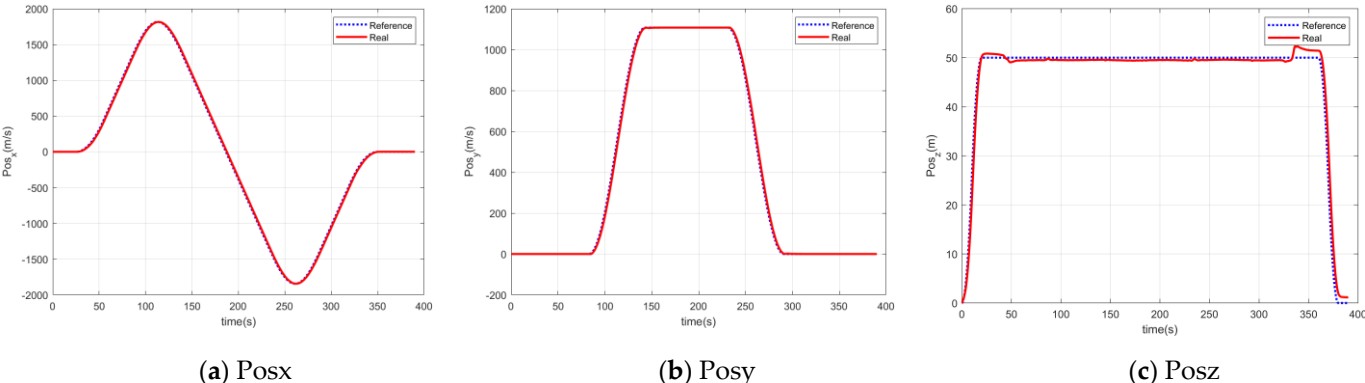

(**a**) Posx    (**b**) Posy    (**c**) Posz

**Figure 19.** Flight state response curve of position in position mode control simulation.

## 6. Conclusions

This article introduces the system design, dynamics modeling, controller design and flight simulation of a tilt-rotor VTOL UAV. Using basic aerodynamic theory, a complete dynamic model of the aerodynamic effects of a tilt-rotor UAV is derived. In addition, we also developed a hierarchical fusion control system based on ADRC and applied it to the full-mode simulated flight of UAVs. The simulation results demonstrate the aircraft's capability to autonomously execute essential flight maneuvers, including vertical takeoff and landing, hovering, forward transition, horizontal flight, and backward transition, and both the velocity loop and position loop can respond well to control instructions under external disturbances. These simulations show that the developed aircraft can complete autonomous flight tasks in complex environments. Future work will focus on flight controller parameter optimization, complete aircraft hardware-in-the-loop simulation, and outdoor flight tests.

**Author Contributions:** Conceptualization, Z.L.; methodology, Z.L.; software, Z.L.; validation, Z.L. and Z.X.; formal analysis, Z.L., Z.X. and G.W.; investigation, Z.L.; resources, L.F.; data curation, Z.L.; writing—original draft preparation, Z.L.; writing—review and editing, G.W. and Z.X.; visualization, Z.L. and Z.X.; supervision, L.F., G.W. and Z.X.; project administration, L.F., G.W. and Z.X.; funding acquisition, L.F. All authors have read and agreed to the published version of the manuscript.

**Funding:** This work is supported by the Intelligent Aerospace System Team of Zhejiang Provincial Leading innovative Teams Program, Science and Technology Department of Zhejiang Province (Grant No. 2022R01003).

**Data Availability Statement:** Data are contained within the article.

**Conflicts of Interest:** The authors declare no conflict of interest.

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
