# Peer review of "Design, Modeling, and Control of a Composite Tilt-Rotor Unmanned Aerial Vehicle"

_drones, doi:10.3390/drones8030102_

Round 1

Reviewer 1 Report

Comments and Suggestions for Authors

The paper designs a tilt-rotor eVTOL with two tiltable propellers and four fixed propellers for vertical lift and constructs a model that takes the degradation of motor thrust and torque in the presence of forward speeds into consideration. A fusion ADRC control strategy is proposed for this type of tilt-rotor UAV. The topic of this manuscript is interesting. The content is reasonable. However, the superiority of the configuration of the proposed eVTOL is not clear, and some issues still need to be improved:

1, Only 2 of 6 rotors can tilt to generate forward thrust, which means the other four rotors will become the deadweight in the fixed-wing model.

2. Mounting the fixed-axis rotors directly behind the tilt rotors will cause thrust loss issues.

3. Why are bevel gears used in the transmission mechanism? It seems that using servos to directly drive the cylindrical gears will reduce weight of the transmission mechanism.

4. The designed control algorithm is merely a direct application of the ADRC, with limited improvements in the control design or theoretical contributions.

5. The designed eVTOL has a high level of completion. It is suggested to add experimental results.

6, It is claimed that ESO can simultaneously estimate system internal and total external disturbances. However, this characteristic is not reflected in the simulation results. It is suggested to include anti-interference experimental results to enhance the logical coherence of the entire paper.

7. What is the meaning of Pos_sp in Fig. 7? 

Comments on the Quality of English Language

The author should proofread the manuscript carefully.

Author Response

We want to thank sincerely the reviewer for his/her constructive comments regarding our paper. Our response to each comment is given below. The marked sections in the revised word (pdf)are marked by red.

Comments: “The paper designs a tilt-rotor eVTOL with two tiltable propellers and four fixed propellers for vertical lift and constructs a model that takes the degradation of motor thrust and torque in the presence of forward speeds into consideration. A fusion ADRC control strategy is proposed for this type of tilt-rotor UAV. The topic of this manuscript is interesting. The content is reasonable. However, the superiority of the configuration of the proposed eVTOL is not clear, and some issues still need to be improved”

We also appreciate your clear and detailed feedback and hope that the explanation has fully addressed all of your concerns. In the remainder of this letter, we discuss each of your comments individually along with our corresponding responses. To facilitate this discussion, we first retype your comments in italic font and then present our responses to the comments.

Comment 1:

Only 2 of 6 rotors can tilt to generate forward thrust, which means the other four rotors will become the deadweight in the fixed-wing model.

Response 1:

Thank you very much for your feedback. It should be explained that the original intention of our designed aircraft is:

(1) Compared to standard vertical takeoff and landing aircraft, the target aircraft utilizes a forward tilt mechanism to provide some thrust support for the fixed wing mode propulsion blades in rotor mode, improving the power system utilization efficiency and maximum takeoff weight of VTOL during hovering;

(2) The target aircraft has a greater safety margin in the mode transition process compared to traditional tiltrotor and fully tiltrotor quadcopters. As can be seen from the following figure, due to the presence of four fixed rotors at the rear, the forward tilt motor can still achieve force and torque balance in the pitch dimension of the aircraft at a lower tilt angle during low-speed flight. Compared to traditional tiltrotor, it expands the tilt corridor;

Figure 1: Transition Tilting Corridor of Our Aircraft

Figure 2: Transition Tilting Corridor of Traditional Tilting Twin Rotors

(3) From the perspective of redundant control, using multiple actuators in rotor mode can effectively reduce the probability of a single actuator failure leading to the failure of the entire system. Obviously, using a distributed six rotor system is safer than traditional twin rotor, three rotor, and four rotor aircraft. However, for tiltrotor aircraft with fixed blade angles, It is worth considering whether all power systems need to be used for tilting. On the one hand, increasing the propeller speed and using fixed wing propellers in fixed wing mode are more critical than simply increasing the number of propellers working in fixed wing mode. On the other hand, this contradicts the concept of distributed redundant control, making the excess rotor become the weight of the fixed wing model.

Finally, to highlight the contribution of the article, we extract our innovative points in the first chapter (lines 120-132) of the new manuscript and explain the design purpose of this configuration in Section 2.1 (lines 181-189).

However, based on the current results, the target aircraft at this stage has the disadvantage that four rotors in fixed wing mode will become the self weight of fixed wing models. In subsequent research, our laboratory is designing a vertical takeoff and landing drone with 6-rotor fully tilted blades, planned to be equipped with periodic variable pitch propellers, which can effectively solve the efficiency problem of propellers in rotor and fixed wing modes, while avoiding the other four rotors becoming self weight in the fixed wing model.

Comment 2:

Mounting the fixed-axis rotors directly behind the tilt rotors will cause thrust loss issues.

Response 2:

Thank you very much for your feedback. Installing the fixed shaft rotor directly behind the tilting rotor can lead to thrust loss issues. At the same time, accurately modeling the interaction between propellers is a challenging task. In order to reduce the interaction between propellers in transition and fixed wing modes, we have redesigned the control parameters of the "Vertical take off landing fusion controller" module, It is stipulated that  is 16.5m/s, which means that when the forward flight speed is less than 16.5m/s, the tilting rotor does not deflect but remains vertical. When the current flight speed is greater than 16.5m/s, the tilting rotor quickly deflects to the horizontal position. The relevant formula is:

Through re simulation, it can be found that after repeating the speed simulation experiment in section 5.2, the aircraft can still fly stably after changing the value of  in the tilt controller. However, during the forward transition period, the deviation occurs between 25.6 and 27.3 seconds. At 27.1 seconds, the speed of the four motors in the rear decreases to 0 and the blades are locked in a position parallel to the fuselage according to the program, reducing resistance; At the same time, according to calculations, when the tilt angle is 51.3 °, that is, the wake of the front propeller will start sweeping to the 3rd to 4th propellers, and this time is 26.4s, which means that the true duration of the influence of the front propeller is 0.7s; In the backward conversion, the time for deflection is from 40.1 to 42.3 seconds. Currently, the speed of the four rear motors is 0, and the blades are locked in a position parallel to the body according to the program.

Figure 3 Changes in tilt motor angle over time

Figure 4 Changes in motor speed over time

In the simulation of the new version of the manuscript, we adopted the latest control scheme and conducted re-simulation (see Section 5.2/5.3 for details, lines 540-541)

In summary, after the modification of the program, we have reduced the problem of thrust loss caused by tilting the rotor. In future work, we will consider rearranging the positions of the six propellers, especially separating the positions of propellers 1 to 2 and 3 to 6 projected on the yoz plane to avoid mutual influence between propellers during the tilting process.

Comment 3:

Why are bevel gears used in the transmission mechanism? It seems that using servos to directly drive the cylindrical gears will reduce weight of the transmission mechanism.

Response 3:

Thank you very much for your question. Here is our explanation:

(1) We mainly use bevel gears to change the direction of the motor output shaft. If cylindrical gears are used, the motor needs to be installed on the side of the tilting mechanism, which increases the lateral volume of the tilting mechanism. Therefore, bevel gear transmission was adopted to change the direction of the shaft system transmission, and the motor can be installed at the longitudinal end of the tilting mechanism, which is easy to be wrapped by the power short compartment skin, forming a slender cylindrical shape, and reducing aerodynamic resistance. (lines 191-197 of the new version manuscript)

(2) In addition, according to our design goal, the tilting mechanism in front should achieve a specified angle of inclination within the range of -45 ° to 135 ° and achieve a 1:10 reduction ratio to prevent the tilting mechanism from slipping around the self-rotating axis under complex force conditions. So within the tilt range of -45 ° to 135 °, the tilt motor rotates at least 1800 deg. As far as we know, most PWM and CAN servos currently available on the market do not support specified angle control in a large angle range, while CAN motors can meet the above requirements and have similar quality as servos. Therefore, we choose CAN motors. (lines 200-206 of the new version manuscript)

Comment 4:

The designed control algorithm is merely a direct application of the ADRC, with limited improvements in the control design or theoretical contributions.

Response 4:

Thank you very much for your suggestion. The theme of this article is to conduct overall design, modeling, and control simulation work on a new type of composite tilt VTOL UAV, focusing on accurate dynamic modeling of the target aircraft and achieving full process flight control through robust control methods, in preparation for subsequent physical hardware in the loop simulation and flight tests. Due to the overall weight exceeding 30KG, it belongs to medium to large tilt unmanned aerial vehicles. In order to improve the success rate of physical flight tests and accelerate the flight test process, we have adopted ADRC control on the basis of traditional hybrid PID to improve the anti-interference ability of the controller, and plan to conduct tests in the near future. In the future, we will conduct innovative theoretical research on the control methods of tiltrotor based on this work, and achieve innovative theoretical control simulation and experiments on scaled small tiltrotor.

Comment 5:

The designed eVTOL has a high level of completion. It is suggested to add experimental results.

Response 5:

Thank you very much for your suggestion. Our goal is to apply the content of this paper to practice and achieve full profile flight control of the target compound tilt aircraft. We are currently working towards this direction as well.

Comment 6

It is claimed that ESO can simultaneously estimate system internal and total external disturbances. However, this characteristic is not reflected in the simulation results. It is suggested to include anti-interference experimental results to enhance the logical coherence of the entire paper.

Response 6:

Thank you very much for your suggestion. In the new version of the manuscript, we have considered the factors of sensor measurement error and external wind disturbance. Among them, we use white noise with a sampling rate of 0.2 and an amplitude of 0.05 to simulate the IMU sensor observation error, and add it to the attitude measurement values; Add white noise with a sampling rate of 0.2 and an amplitude of 0.1 to the measured velocity to simulate GPS sensor measurement errors. In addition, external wind disturbances can interfere with the flight of UAVs, and we use sine wave signals to simulate changes in wind disturbances. Add an external wind disturbance with a wind speed of vf=2sin (0.5t) to the body's three-axis coordinate system.

The simulation results are presented in sections 5.2 and 5.3 of the new manuscript, and the system can still stably follow the command angle in the presence of external interference and sensor measurement errors.

Comment 7

What is the meaning of Pos_sp in Fig. 7? 

Response 7:

We apologize for any confusion caused by the unclear explanation of the previous images, where Pos_sp is the expected position coordinate. To avoid ambiguity, we have replaced Pos_sp with Pos_d (desire) in the new version of the manuscript.

We would like to take this opportunity to thank you for all your time involved and this great opportunity for us to improve the manuscript. We hope you will find this revised version satisfactory.

Sincerely,

The Authors

Reviewer 2 Report

Comments and Suggestions for Authors

The manuscript describes the design, modeling and control of a tilt-rotor UAV. Simulations are performed as a matter of validation.

The claimed contribution of the paper is the introduction of a new 6-rotor tiltrotor vehicle. However, it is unclear in which ways the vehicle is actually different from other tiltrotors in the literature (except from the precise number of rotors). This concern is reinforced by the fact that the authors adopt the methods for modeling and control from the literature. I think a 'design paper' like this is more suited for a conference than a journal, where a clear scientific contribution is required.

Connected to this issue, few references have been used, of which some are websites. The work should be placed in the context of the relevant work done in this area, and there are much more relevant papers than are cited.

The paper is well-structured, but much longer than needed. Some sections are not superfluous, and contain many repetitions. At the same time, the paper does not go into detail on the methods employed. Often it is just stated what the design outcome is (e.g. which airfoil profile is used), without explaining how this is obtained, greatly reducing the value of the paper. The same holds for the modeling and control sections.

The interpolation between commands from the hover and fixed wing controllers may work in some cases, but there is no guarantee that it will always work.  The argumentation in the paper doesn't go further than "it has been found to work well", which is not sufficient, given that alternatives exist.

It is good that simulations have been preformed, but the results look like the controller has very high gain. Such a controller would be very susceptible to vibrations and noise, which are clearly not modeled. I'm curious if it still performs the same if these are included. Please also have a look at the literature, papers that include real experiments don't show 0.1 degree accuracy results.

Please also consider the comments in the PDF. Where possible, figures should be vector graphics.

Comments on the Quality of English Language

The paper contains many errors, some of which I have highlighted in the attached PDF. Some of these errors could have been prevented with a simple grammar checker.

Some sentences cannot be understood:

"The influence of the induced speed brought by the propeller disk and the nor-68 mal speed brought by the tilt on the resultant angle of attack."

"Zheng Chen et al. [10] designed a full-mode flight control law for tilt-rotor aircraft based on the incremental nonlinear dynamic inverse method based on the characteristics of high model accuracy and poor robustness of nonlinear dynamic inverse control." - INDI is not based on high model accuracy of NDI, it alleviates the need for a very accurate model.

Author Response

We want to thank sincerely the reviewer for his/her constructive comments regarding our paper. Our response to each comment is given below. The marked sections in the revised word (pdf)are  marked by red.

Comments: “The manuscript describes the design, modeling and control of a tilt-rotor UAV. Simulations are performed as a matter of validation.”

We also appreciate your clear and detailed feedback and hope that the explanation has fully addressed all of your concerns. In the remainder of this letter, we discuss each of your comments individually along with our corresponding responses. To facilitate this discussion, we first retype your comments in italic font and then present our responses to the comments.

Comment 1:

The claimed contribution of the paper is the introduction of a new 6-rotor tiltrotor vehicle. However, it is unclear in which ways the vehicle is actually different from other tiltrotors in the literature (except from the precise number of rotors). This concern is reinforced by the fact that the authors adopt the methods for modeling and control from the literature. I think a 'design paper' like this is more suited for a conference than a journal, where a clear scientific contribution is required.

Response 1:

Thank you very much for your feedback. It should be explained that the purpose of our designed aircraft is:

  • We hope that the target aircraft can use the forward tilt mechanism to provide some thrust support for the fixed wing mode propeller in rotor mode, compared to the standard vertical takeoff and landing aircraft, to improve the power system utilization efficiency and maximum takeoff weight of VTOL during hovering;
  • The target aircraft has a greater safety margin in the mode transition process compared to traditional tiltrotor and fully tiltrotor quadcopters. As can be seen from the following figure, due to the presence of four fixed rotors at the rear, the forward tilt motor can still achieve force and torque balance in the pitch dimension of the aircraft at a lower tilt angle during low-speed flight. Compared to traditional tiltrotor, it expands the tilt corridor;
  • From the perspective of redundant control, the use of multiple actuators in a distributed six rotor system in rotor mode can effectively reduce the probability of system failure caused by a single actuator failure. Obviously, using a distributed six rotor system is safer than traditional twin rotor, three rotor, and four rotor aircraft;

Figure 1: Transition Tilting Corridor of Our Aircraft

Figure 2: Transition Tilting Corridor of Traditional Tilting Twin Rotors [1]

In addition, the main theme of this article is the overall design, modeling, and control simulation of a composite tilt VTOLUAV, focusing on accurate dynamic modeling of the target aircraft and achieving full process flight control through robust control methods, preparing for subsequent physical hardware in the loop simulation and flight tests. Due to the overall weight exceeding 30KG, it belongs to medium to large tilt unmanned aerial vehicles. In order to improve the success rate of physical flight tests and accelerate the flight test process, we have adopted ADRC control on the basis of traditional hybrid PID to improve the anti-interference ability of the controller, and plan to conduct tests in the near future.

In conclusion, we believe that the article is somewhat innovative. In the future, we will conduct innovative theoretical research on the control methods of tiltrotor based on this work, and achieve innovative theoretical control simulation and experiments on scaled small tiltrotor. In the new version of the manuscript, we have added relevant explanations in the introduction (lines 120-132) and chapter two (lines 181-189).

Comment 2:

Connected to this issue, few references have been used, of which some are websites. The work should be placed in the context of the relevant work done in this area, and there are much more relevant papers than are cited.

Response 2:

Thank you very much for your feedback. We agree with your suggestion that the reference materials used were not rich enough. We have added 11 additional articles in the new manuscript (mainly the first introduction), which mainly involve the latest articles on tilt rotor control. At the same time, we have added citations on the methods used in dynamic and control modeling in the articles to the references.

Comment 3:

The paper is well-structured, but much longer than needed. Some sections are not superfluous, and contain many repetitions. At the same time, the paper does not go into detail on the methods employed. Often it is just stated what the design outcome is (e.g. which airfoil profile is used), without explaining how this is obtained, greatly reducing the value of the paper. The same holds for the modeling and control sections.

 Response 3:

Thank you very much for your comments and suggestions on this article. We apologize for any confusion caused by unclear explanations in the previous article. In the new manuscript, we will make partial revisions to the second and third chapters of the article according to your suggestions.

In Chapter 2, we rephrased the section on aircraft design, including wing profile, and added design reasons and references (see lines 150-161 for details) based on the design results. We also made significant simplifications to the aircraft mode transition and mechanism design sections (see lines 173-178 and 190-197 for details). In Chapter 3, we extensively simplified the definition of the coordinate system in Section 3.1 and added deYoung's empirical formula and references to the Vienna and Corrigan model in Sections 3.2 and 3.3.

Comment 4:

The interpolation between commands from the hover and fixed wing controllers may work in some cases, but there is no guarantee that it will always work.  The argumentation in the paper doesn't go further than "it has been found to work well", which is not sufficient, given that alternatives exist.

Response 4:

Thank you very much for your suggestion. Based on simulation results, it has been shown that using F-ADRC can effectively achieve full flight profile flight control of the target aircraft. At the same time, we are preparing to use hardware in the loop simulation to verify flight stability.

As you mentioned, there are currently unified control methods such as Model Predictive Control (MPC) and Deep Reinforcement Learning that have been used to control tiltrotor unmanned aerial vehicles. However, it should be noted that due to the overall weight of our research object exceeding 30KG, which belongs to medium to large tiltrotor unmanned aerial vehicles, aerodynamic disturbances and model modeling accuracy have a significant impact on control effectiveness. As far as the author knows, aircraft of this mass scale and above mostly use linear interpolation for transition flight. Therefore, to improve the success rate of physical flight tests and accelerate the flight test process, we use ADRC control based on traditional hybrid PID to improve the controller's anti-interference ability, and plan to conduct experiments soon.

In the future, we will conduct innovative theoretical research on the control methods of tiltrotor based on this work, and achieve innovative theoretical control simulation and experiments on scaled small tiltrotor.

Comment 5:

It is good that simulations have been preformed, but the results look like the controller has very high gain. Such a controller would be very susceptible to vibrations and noise, which are clearly not modeled. I'm curious if it still performs the same if these are included. Please also have a look at the literature, papers that include real experiments don't show 0.1 degree accuracy results.

Response 5:

Thank you very much for your suggestion. After discussion and simulation, we have found that as you mentioned, our initial version of the controller had a high parameter gain and was prone to divergence in the presence of noise interference. In the new version of the manuscript, we have re selected control parameters with lower gains and conducted re simulation.

In terms of vibration and noise modeling, we consider factors such as sensor measurement errors and external wind disturbances. Among them, we use white noise with a sampling frequency of 0.1 and an amplitude of 0.1 to simulate IMU sensor observation errors and add them to the attitude measurement values[2]; Add white noise with a sampling frequency of 0.1 and an amplitude of 0.05 to the measured velocity to simulate GPS sensor measurement errors. In addition, external wind disturbances can interfere with the flight of UAVs, and we use sine wave signals to simulate changes in wind disturbances. Add an external wind disturbance with a wind speed of Vf=0.5sin (0.5t) to the body's three-axis coordinate system[3].

Through simulation, under a certain range of disturbances, the aircraft can stably track the command angle and command speed. At the same time, due to measurement noise and external disturbances, there is a deviation of 0.05-0.5deg between the true value of the angle and the reference value, with a maximum error of 1.1deg.

However, there are still differences between this and the actual experimental results. Accurately modeling the flight environment model, actuator model, sensor model, and structural flexibility is also a very challenging task. In the future, we will conduct flight tests and collect flight data with protective measures in place.

Next, we will respond to the questions in the PDF which you uploaded:

Comment 6:

The main wing and canard wing both adopt the Beckley wing shape What is this?

Response 6:

We apologize for any confusion caused by unclear explanations in the previous article. We have provided explanations in the new version of the manuscript, please refer to lines 154-156 for details.

Comment 7:

with the canard wing adopting a flat wing shape and a designed installation angle of attack of 1 deg with the fuselage. These details are useless without the design steps. Why was this angle chosen?

Response 7

We apologize for the confusion caused by the unclear explanation of the previous article. In the new version of the manuscript, we have re described this paragraph and explained the design reasons. Please refer to lines 156-161 for details.

Comment 8:

The tail design of the wing is equipped with a wingtip small wing to reduce the intensity of the wingtip vortex and reduce the induced drag of the wing. this is called a winglet

Response 8:

Thank you very much for your correction. We have made the necessary changes in the new manuscript.

Comment 9:

“Add a pair of downward facing V-shaped tail” this sounds like an imperative

Response 9:

Thank you very much for pointing out the issue. In the new manuscript, we have rephrased this paragraph. Please refer to lines 158-161 for details.

Comment 10:

“ In addition to tilting together with the tilting mechanism, the independent canard tilting part can also tilt independently by -15~15° relative to the main fuselage in fixed-wing mode (the tilting mechanism is locked in the horizontal position)”

This is unclear, it should be able to tilt 90 degrees, so 15 degrees should be inherently possible.

Response 10:

We apologize for any confusion caused by the unclear explanation of the previous article. In the new version of the manuscript, we will provide a new description of this paragraph. Please refer to lines 164-167 for details.

Comment 11:

“the four tilt rotors on the main wing stop rotating and adjust the blades to be parallel to the fuselage”

 These don't tilt as far as I understand.

Response 11:

We apologize for the confusion caused by the unclear explanation in the previous article. What we want to express is that the motor speed drops to 0 (stops running). In the new version of the manuscript, we will provide a new description of this paragraph.

Comment 12:

This section contains many redundant sentences / repetitions.

Response 12:

Thank you very much for pointing out the issue. In the new manuscript, we have rephrased and deleted this paragraph. Please refer to lines 173-178 for details.

Comment 13:

“At present, tilting wings generally use linear actuators, servos, or servo motors.” I don't think so? And no reference is given at all!

Response 13:

We apologize for any confusion caused by the unclear explanation of the previous article, and the citation in this section is from Chapter 2 of [4]. After discussion, we believe that this paragraph does not contribute much to the article and were deleted in the new version of the manuscript.

Comment 14

“The standard PWM servo system only has a nominal 360 degree rotation range” So? This should be more than enough, you only need ~90 degrees.

Response 14

We apologize for any confusion caused by the unclear explanation in the previous article. It should be noted that due to the 1:10 reduction ratio during the output of the tilting motor to the tilting mechanism, the tilting motor rotates at least 1800deg within the tilting range of -45 deg to 135 deg. After discussion, we believe that this paragraph does not contribute much to the article. Therefore, we delete it in the new version of the manuscript and rephrase it. Please refer to lines 200-206 for details.

Comment 15

“equation” This model does not consider inflow and will be inaccurate in forward flight; the predicted thrust will be too high.

Response 15

I'm sorry for not explaining it clearly here. It should be noted that we have already taken into account the impact of inflow on propeller thrust in the article. Please refer to line 260 for details. In the new manuscript, we use  instead of  to avoid ambiguity.

Comment 16

“We use de-Young's empirical formula to estimate these normal forces, which are calculated as follows.” reference?

Response 16

We are very sorry that there is no cited literature here. We have corrected it in the new manuscript.

Comment 17

Is this significant? The rotor inertia is probably very small compared to the vehicle's inertia. Also, for pitch changes, if xhi_1 equals xhi_2 the terms cancel if the omegas are equal. Also, stricly speaking I_rotor is time-varying here, depending on the propeller rotation.

Response 17

Thank you very much for your feedback. After multiple discussions, we have found that the contribution of this part of the torque to the overall torque is indeed very small, and the I_rotor here varies over time, depending on the rotation of the propeller, making it difficult to accurately model. In the new version of the manuscript and simulation program, we have removed option . And make changes to the relevant parts of the article.

Comment 18

“Among them, v represents the UAV's speed in the inertial frame, and w represents the wind speed expressed in the earth frame system.” The frames have to be the same if you are subtracting them.

Response 18

Thank you very much for pointing out the error here. In the new manuscript, we have included coordinate system one.

Comment 19

“We use a finite length rectangular wing model developed by Tangler and Ostowari based on experimental data and Viterna and Corrigan's model. The lift and drag coefficients after stall are given by” reference?

Response 19

We are very sorry that there is no cited literature here. We have corrected it in the new manuscript.

Comment 20

“There are three flight states for tiltrotor aircraft: rotor mode, fixed wing mode” is it a flight state or a mode?

Response 20

I'm very sorry for not explaining it clearly here. We are referring to the flight mode, which has been corrected in the new manuscript. At the same time, we have changed the "rotor mode" to "rotary wing mode"

Comment 21

while the rotor position controller comes from I think you mean rotorcraft, or the hover mode. This wording suggest you're talking about the tilting rotors.

Response 21

I'm very sorry for not explaining it clearly here. We are referring to "hover mode" or "rotary wing mode" in the new manuscript, which we have corrected. Thank you for pointing it out.

Comment 22

These two position controllers have been implemented on the existing framework of Pixhawk autonomous driving software PX4, pixhawk is the hardware,flying.

Response 22

I'm very sorry for not explaining it clearly here. We have corrected it in the new manuscript.

Comment 23

“The horizontal dynamics of a multi-rotor aircraft (in the fuselage) are decoupled from the vertical dynamics because thrust only acts in the a direction.” what is meant by this?

Response 23

I'm very sorry for not explaining clearly here. We are referring to "in the body coordinate system". In the new manuscript, we have reorganized the language and made corrections. We apologize for any confusion caused by unclear explanations in the previous article.

Comment 24

“be reduced to 11 degrees of freedom: two elevators have the same tilt angle in the same direction, two ailerons have the same tilt angle in the opposite direction, and the rudder has the same tilt angle in the same direction.” Why is this done? To reduce the computation load?

Response 24

I'm sorry for not explaining it clearly here. Reducing the number of actuators from 14 to 11 can reduce the dimension of the mixing matrix, thereby reducing the complexity of control. We have reorganized the language for description in the new manuscript.

Comment 25

Confusing nomenclature with the modeled moments (e.g. M_aero)

I'm missing where these matrices come from. Are they computed with a pseudo inverse?

Response 25

We are very sorry for the writing issue here. In the new manuscript, we have corrected the formula here (lines 491-497). The acquisition of mixed control matrix can refer to [5]

Comment 26

“The oscilloscope is also placed where needed to display the flight status in real-time.”

This is a well known feature of simulink and completely irrelevant to mention here.

Response 26

Thank you very much for pointing out the issue here. We have deleted it in the new manuscript.

Comment 27

This is superfluous since the references are provided in the figures.

Response 27

Thank you very much for pointing out the issue here. We have deleted it in the new manuscript.

Comment 28

As the deceleration increases, the pitch direction angle gradually increases to 0.15deg, and then the pitch direction rapidly decreases. The maximum tracking error of the pitch channel as a whole is within 0.01 °.” This is not realistic, especially for a controller that doesn't include any feed-forward commands, and can only be achieved in simulation, as can be confirmed by investigating the literature.

Response 28

Thank you very much for pointing out the issue here. In the new manuscript, we have added external disturbances and sensor measurement errors to the simulation environment. Through further simulation, we have found a deviation of 0.05-0.5deg between the actual angle value and the reference value, with a maximum error of 1.1deg. Please refer to sections 5.2 and 5.3 of the beginner's manuscript for details. However, there are still differences between this and the actual experimental results. Accurately modeling the flight environment model, actuator model, sensor model, and structural flexibility is also a very challenging task. In the future, we will conduct flight tests and collect flight data with protective measures in place.

Comment 28

“The maximum error of the three axis velocity tracking occurs during the transition from rotor to fixed wing and from fixed wing to rotor.” any reasoning on why this is?

Response 28

Sorry for not providing an explanation here. After analysis, we believe that on the one hand, the tiltrotor drone is a variable structure process with strong coupling characteristics during the transition process. On the other hand, there may be significant changes in the control variables between the fixed wing and rotor controllers during the switching process. The relevant explanations have been added to the new manuscript.

Comment 29

Please also consider the comments in the PDF. Where possible, figures should be vector graphics.

Response 29

We thank you again for your detailed review of the article content. In the new manuscript, we have bolded the vector symbols in the formulas, and we have also numbered all the formulas in the article.

We would like to take this opportunity to thank you for all your time involved and this great opportunity for us to improve the manuscript. We hope you will find this revised version satisfactory.

Sincerely,

The Authors

References

  • Yun, C.; Research on Mathematical Modeling Method for Tilt Rotor Aircraft Flight Dynamics. Nanjing University of Aeronautics and Astronautics, 2012.
  • Xu, Z.; Fan, L.; Qiu, W.; Wen, G.; He, Y. A Robust Disturbance-Rejection Controller Using Model Predictive Control for Quadrotor UA V in Tracking Aggressive Trajectory. Drones 2023, 7, 557.
  • Shen, S.; Xu, J.; Chen, P.; Xia, Q. Adaptive Neural Network Extended State Observer-Based Finite-Time Convergent Sliding Mode Control for a Quad Tiltrotor UAV. IEEE Transactions on Aerospace and Electronic Systems, 2023, PP. 1-14.
  • North D D , Busan R C , Howland G .Design and Fabrication of the Langley Aerodrome No. 8 - Distributed Electric Propulsion VTOL Testbed[C]//AIAA SciTech Conference.2020.
  • Spannagl, L.; Ducard, G.; Control Allocation for an Unmanned Hybrid Aerial Vehicle. 2020 28th Mediterranean Conference on Control and Automation (MED), 2020, 709-714.

Reviewer 3 Report

Comments and Suggestions for Authors

Dear authors,

Your paper addresses an important and challenging topic in the field of UAVs  flight dinamics. The paper is of good quality, well structured and correctly grounded, but I suggest (suggestion, not mandatory) that it can be improved in terms of: a) a) the expansion of the conclusions section (possibly by transferring some information already presented in the previous results-section); b) the repositionning of the appendix inside the paper's content, the figures being more concludent for the simulations section.

However, this paper is only the first step; as you already declared, controller's optimization and  experimental validation are mandatory.

Author Response

Dear Reviewers,

Thank you very much for your time involved in reviewing the manuscript and your very encouraging comments on the merits.

Comments:

“Your paper addresses an important and challenging topic in the field of UAVs  flight dinamics. The paper is of good quality, well structured and correctly grounded, but I suggest (suggestion, not mandatory) that it can be improved in terms of: a) the expansion of the conclusions section (possibly by transferring some information already presented in the previous results-section); b) the repositionning of the appendix inside the paper's content, the figures being more concludent for the simulations section.

However, this paper is only the first step; as you already declared, controller's optimization and  experimental validation are mandatory.”

We also appreciate your clear and detailed feedback and hope that the explanation has fully addressed all of your concerns. In the remainder of this letter, we discuss each of your comments individually along with our corresponding responses. To facilitate this discussion, we first retype your comments in italic font and then present our responses to the comments.

Comment 1:

the expansion of the conclusions section (possibly by transferring some information already presented in the previous results-section);

Response 1:

Thank you very much for your feedback. In the new version of the manuscript, we have expanded the conclusion section and added the performance of the controller in the presence of external disturbances and measurement errors. Please refer to sections 5.2 and 5.3 for details

Comment 2:

the repositionning of the appendix inside the paper's content, the figures being more concludent for the simulations section.

Response 2:

Thank you very much for your feedback. After discussion, we agree with your point that these data are indeed more important for the simulation section. In the new version of the manuscript, we will move the aerodynamic data results in the appendix to section 5.1

Comment 3:

 However, this paper is only the first step; as you already declared, controller's optimization and experimental validation are mandatory.

Response 3:

Thank you very much for your feedback. Our goal is to apply the content of this paper to practice and achieve the full profile flight control flight test of the target compound tilt aircraft. We plan to optimize and improve the controller based on this work, and we are currently working towards this direction.

We would like to take this opportunity to thank you for all your time involved and this great opportunity for us to improve the manuscript. We hope you will find this revised version satisfactory.

Sincerely,

The Authors

Round 2

Reviewer 1 Report

Comments and Suggestions for Authors

I am satisfied with the revised version. Considering the visualization block in your simulation environment, it is suggested to include the link of the simulation video to better demonstrate the flight performance of the tilt-rotor eVTOL system.

Author Response

This document contains a set of replies to the comments made by the reviewers for the paper drones-2845039, titled “Design, modeling, and control of a composite Tilt-rotor Unmanned Aerial Vehicle”.

We want to sincerely thank you and the reviewers for evaluating our paper. The feedback received helped us to greatly improve our work. We have now prepared a revised version of our paper that addresses all issues raised by the reviewers.

Cordially,

Zhuang Liang and Li Fan

  • Response to Reviewer 1

We want to thank sincerely the reviewer for his/her constructive comments regarding our paper. Our response to each comment is given below.

Comments: “I am satisfied with the revised version. Considering the visualization block in your simulation environment, it is suggested to include the link of the simulation video to better demonstrate the flight performance of the tilt-rotor eVTOL system.”

We appreciate your clear and detailed feedback. We fully agree with your suggestion. In the new version of the manuscript, we have added links to relevant simulation videos in section 5.3(https://www.youtube.com/watch?v=pcORCEPILsA)

Figure 1 Visual simulation in position mode.

We would like to take this opportunity to thank you for all your time involved and this great opportunity for us to improve the manuscript. We hope you will find this revised version satisfactory.

Sincerely,

The Authors
